# Short-term response of *Emiliania huxleyi* growth and morphology to abrupt salinity stress

Rosie M. Sheward[1], Christina Gebühr[1,2], Jörg Bollmann[3], Jens O. Herrle[1,2]

[1]Institute of Geosciences, Goethe-University Frankfurt, Frankfurt am Main, D-60438, Germany
[2]Biodiversity and Climate Research Centre (BIK-F), Frankfurt am Main, D-60325, Germany
[3]Department of Earth Sciences, University of Toronto, Toronto, M5S3B1, Canada

*Correspondence to*: Rosie M. Sheward (sheward@em.uni-frankfurt.de)

**Abstract.** The marine coccolithophore species *Emiliania huxleyi* tolerates a broad range of salinity conditions over its near-
global distribution, including the relatively stable physiochemical conditions of open ocean environments and nearshore environments with dynamic and extreme short-term salinity fluctuations. Previous studies show that salinity impacts the physiology and morphology of *E. huxleyi*, suggesting that salinity stress influences the calcification of this globally important species. However, it remains unclear how rapidly *E. huxleyi* responds to salinity changes and therefore whether *E. huxleyi* morphology is sensitive to short-term, transient salinity events (such as occur on meteorological timescales) in addition to
longer duration salinity changes. Here, we investigate the real-time growth and calcification response of two *E. huxleyi* strains isolated from shelf-sea environments to the abrupt onset of hyposaline and hypersaline conditions over a time period of 156 h (6.5 days). Morphological responses in the size of the cell covering (coccosphere) and the calcium carbonate plates (coccoliths) that form the coccosphere occurred as rapidly as 24-48 h following the abrupt onset of salinity 25 (hyposaline) and salinity 45 (hypersaline) conditions. Generally, cells tended towards smaller coccospheres (-24%) with smaller coccoliths (-7 to -11%)
and reduced calcification under hyposaline conditions whereas cells growing under hypersaline conditions had either relatively stable coccosphere and coccolith sizes (Mediterranean strain RCC1232) or larger coccospheres (+35%) with larger coccoliths (+13%) and increased calcification (Norwegian strain PLYB11). This short-term response is consistent with reported coccolith size trends with salinity over longer durations of low and high salinity exposure in culture and under natural salinity gradients. The coccosphere size response of PLYB11 to salinity stress was greater in magnitude than observed in RCC1232 but occurred
after a longer duration of exposure (96-128 h) to the new salinity conditions compared to RCC1232. In both strains, coccosphere size changes were larger and occurred more rapidly than changes in coccolith size, which tended to occur more gradually over the course of the experiments. Variability in the magnitude and timing of rapid morphological responses to short-term salinity stress between these two strains supports previous suggestions that the response of *E. huxleyi* to salinity stress is strain specific. At the start of the experiments, the light condition was also switched from a light: dark cycle to
continuous light with the aim of desynchronising cell division. As cell density and mean cell size data sampled every 4 h showed regular periodicity under all salinity conditions, the cell division cycle retained its entrainment to pre-experiment light: dark conditions for the entire experiment duration. Extended acclimation periods to continuous light are therefore advisable

for *E. huxleyi* to ensure successful desynchronisation of the cell division cycle. When working with phased or synchronised populations, data should be compared between samples taken from the same phase of the cell division cycle to avoid artificially 35 distorting the magnitude or even direction of physiological or (bio)geochemical response to the environmental stressor.

## 1 Introduction

Shifts in ocean salinity are an indirect effect of climate change, reflecting changes in the hydrological cycle on seasonal to multi-annual timescales (precipitation, evaporation, and river runoff) and dynamics in ocean circulation patterns and the cryosphere on seasonal to multi-decadal timescales (e.g., Durack et al., 2012; Westra et al., 2014; Haumann et al., 2016; 40 Lenderink and Van Meijgaard, 2008; Yu et al., 2021). Observational records indicate that open-ocean regions of higher salinity have become increasingly saline whilst regions of lower salinity have become fresher (i.e., an amplification of 5-8% or up to ±0.05 pss per decade since 1950; Durack et al., 2012; Zika et al., 2018), enhancing sub-surface salinity in the subtropics and freshening the tropics and sub-polar/polar regions (Durack and Wijffels, 2010; Fox-Kemper et al., 2021). Climate models predict that this trend will be further amplified in future (Fox-Kemper et al., 2021), resulting in spatially variable changes in 45 open-ocean sea surface salinity in the order of an increase/decrease of 0.5 to 1 pss by 2100 relative to the 1985-2014 mean (Röthig et al., 2023). Salinity in coastal areas, continental shelf-seas, marginal seas and (semi-) enclosed basins naturally fluctuate on daily to decadal timescales and salinity trends are more localised and complex, related to a combination of meteorological events, climate-driven changes in the hydrological cycle, and the impact of other local anthropogenic stressors (e.g. land use, vegetation cover, and water management pressures). The complexity of factors influencing salinity in these 50 regions contributes to uncertainty in the magnitude and direction of future salinity trends in coastal areas, marginal seas and (semi-) enclosed basins.

Salinity is a major abiotic factor influencing marine ecosystem structure and function but the impact of salinity on marine organisms has received relatively little attention compared to other climate-related environmental stressors (e.g., Röthig et al., 2023 and references therein). Salinity stress triggers a range of metabolic responses, including internal osmotic and ionic 55 adjustments that can lead to morphological changes, changes to photosynthesis and respiration rates, and biochemical changes such as to osmolyte synthesis (e.g., Kirst, 1990). The response of marine phytoplankton to salinity stress has largely been investigated in coastal and euryhaline species that naturally experience variable salinity conditions, including extreme and/or short-lived salinity events. For example, at one station on the French Atlantic coast, the typical winter salinity range was 21 to 36 with on average four short-lived extreme low salinity events (transient salinity decreases of approx. 2 to >8) each year 60 lasting from a few days to a few weeks (Poppeschi et al., 2021). In the open ocean, extreme salinity anomalies are more typically defined by shifts of 0.2 to 1 pss associated with freshening events, the transit of mesoscale eddies, and atmosphere-ocean dynamics related to El Niño Southern Oscillation, Madden-Julian Oscillation and North Atlantic Oscillation (Liu et al., 2023). Additionally, seasonal monsoons can freshen sea surface salinity by up to 8 pss for several weeks (e.g., at one Bay of Bengal mooring in 2015; Weller et al., 2019) and tropical cyclones can shift salinity by as much as 6 pss for days to weeks

(Xu et al., 2020a). Both coastal and open ocean phytoplankton species therefore experience varying degrees of salinity change over a range of short- and long-term timescales. Laboratory experiments indicate that some marine phytoplankton species are tolerant to salinity change whilst others are not, suggesting that the impact of salinity on phytoplankton physiology may be strain-specific (e.g., Brand, 1984).

One marine phytoplankton species with a demonstrated broad salinity tolerance is the coccolithophore *Emiliania huxleyi*,
which can grow under salinities as low as 11-15 (Brand, 1984; Paasche et al., 1996; Lohbeck et al., 2013) and as high as 38-45 (Bukry, 1974; Brand, 1984; Winter et al., 1979; Fisher and Honjo, 1988; Gebühr et al., 2021; Linge Johnsen et al., 2019; Fielding et al., 2009). *E. huxleyi* has a near-global open ocean distribution (limited presence or absence the very high latitudes, e.g., Winter et al., 2013) but also thrives in the shelf-seas and coastal environments. For example, large blooms of *E. huxleyi* are observed in Norwegian and Patagonian coastal waters and fjords in the summer (e.g., Holligan et al., 1993; Van Der Wal
et al., 1995; Winter et al., 2013; Díaz-Rosas et al., 2021) in addition to shelf-sea blooms and blooms in the open ocean (e.g., Poulton et al., 2013; Tyrrell and Merico, 2004; Poulton et al., 2014; Silkin et al., 2020).

Previous studies have demonstrated that the physiological response of coccolithophores to changing environmental conditions can influence the size of the individual plates (coccoliths) that form an inorganic calcite cell covering (coccosphere) of the coccolithophore (e.g., Faucher et al., 2020; Bollmann, 1997; Bollmann and Herrle, 2007). Laboratory experiments show that
hyper- and hyposaline conditions impact the growth and morphology of many *E. huxleyi* strains (Saruwatari et al., 2016; Green et al., 1998; Paasche et al., 1996; Linge Johnsen et al., 2019; Xu et al., 2020b; Gebühr et al., 2021). Additionally, coccolith size varies systematically along natural sea surface salinity gradients (Bollmann and Herrle, 2007; Bollmann et al., 2009). In light of this morphological sensitivity to salinity conditions, a transfer function relating *E. huxleyi* coccolith size and salinity has been developed to derive paleo-salinity records independently of other geochemical proxies for use in paleoclimate and
paleoceanography research (Herrle et al., 2018; Bollmann et al., 2009; Bollmann and Herrle, 2007). Cellular osmotic adjustments that change cell size have been proposed as the driver of the observed morphological response of *E. huxleyi* coccoliths to salinity stress (Bollmann et al., 2009; Gebühr et al., 2021) but the mechanistic link between salinity and cellular morphology remains unclear. Another open question is how rapidly *E. huxleyi* growth and morphology can respond to salinity changes, which could range from small salinity fluctuations (up to 1; e.g., Liu et al., 2023) over seasonal to annual timescales
in the open ocean or larger magnitude salinity events (increases and decreases of 2 to >10; e.g., Poppeschi et al., 2021; Ridgway, 2007; Weller et al., 2019; Malan et al., 2024; Grodsky et al., 2015) with a rapid onset and short duration that are more likely in coastal and shelf-sea environments.

The aim of this study was to observe the short-term response of *E. huxleyi* growth, morphology, and calcification to an abrupt exposure to low and high salinity conditions by taking measurements for cell concentrations, coccosphere size and coccolith
size every 4 h for 6.5 days. Cultures were simultaneously transitioned from a light: dark cycle to continuous light at the onset of the salinity treatment with the aim that continuous light conditions would desynchronise the cell division cycle by the end of the experiment. Whilst our sampling frequency aimed to identify how quickly morphology responded to salinity stress, an

unintended advantage of our 4 h sampling regime was the real time observation of the response of the cell division cycle to both salinity stress and the onset of continuous light conditions.

## 2 Methods

### 2.1 *Emiliania huxleyi* cultures

The two strains of *E. huxleyi* used in this study were selected to represent an isolate from a lower marine salinity regime contrasted with an isolate from a comparatively high marine salinity regime. Both strains are isolates from coastal regions that may realistically experience larger, transient shifts in salinity compared to open-ocean regions. Strain PLYB11 (Plymouth Culture Collection, UK) is an isolate from the coastline near Bergen, Norway, with a natural seasonal salinity range of 25-33 (Paulino et al., 2018). Strain RCC1232 (Roscoff Culture Collection, France) is a coastal isolate from the Bay of Villefranche-sur-Mer, France. The salinity range in this region of the northwest Mediterranean Sea is approximately 37-38 (Kapsenberg et al., 2017). Stock cultures of both strains were maintained at 15 ºC under 12 h light: 12 h dark conditions at an irradiance of approx. 70 µmol photons $m^{-2}$ $s^{-1}$ and salinity of 35. Cultures were grown in sterile f/2-enriched artificial seawater prepared using a commercial sea salt mixture (Ultramarine, Waterlife Research Industries Ltd., UK) that was dissolved in Milli-Q water with the addition of 0.5 g $L^{-1}$ Tricine to prevent salt precipitation during autoclaving.

### 2.2 Salinity experiments

Cultures of PLYB11 and RCC1232 were grown under three salinity conditions: 25 (low salinity/hyposaline treatment), 35 (control), and 45 (high salinity/hypersaline treatment). The low and high salinity treatments applied here are at the extremes of global open ocean sea surface salinity, which typically range between 29 and 38 (Zweng et al., 2013). However, they encompass the wider range of salinities present some marine settings (e.g., salinities as low as 22-26 in regions of the Arctic Ocean and Baltic Sea and as high as 41 in the Red Sea; e.g., Umbert et al., 2024; Sofianos and Johns, 2017) and the magnitude of extreme salinity change that can occur in coastal regions (e.g., transient decreases in salinity of up to 10 have been reported at some stations and moorings; Poppeschi et al., 2021; Ridgway, 2007; Weller et al., 2019; Grodsky et al., 2015).

The f/2-enriched artificial seawater described above was diluted to the final experimental salinity by the addition of sterilised Milli-Q. To initiate each salinity experiment, triplicate 70 mL polycarbonate flasks with vented lids were prepared with 60 mL of sterile salinity 25, 35, or 45 media and directly inoculated with a small volume of stock culture (35 salinity) to achieve a start concentration of approx. $2x10^5$ cells $mL^{-1}$. Culture flasks were sampled continuously every four hours from the start of the experiment (0 h at 22:00 local time for PLYB11 and 00:00 local time for RCC1232) for 6.5 days (156 h) so that high resolution temporal data could be collected to monitor how rapidly growth and morphology responded to the abrupt change in salinity conditions. At the end of the light phase preceding the start of the experiment (06:00 h to 18:00 h local time), light in the experimental incubator was left on continuously for the duration of the experiment, thereby switching the experimental cultures from a 12 h light, 12 h dark regime to a continuous light regime at 0 h. Continuous light conditions are often used to

desynchronise the cell cycle (e.g., Müller et al., 2008, 2015) and with our 4 h measurement frequency, we were able to additionally track how long the cell division cycle remained entrained to the previous 12:12 light/dark regime by analysing changes in cell concentrations and cell size through the experiments.

## 2.3 Cell concentration and growth

At each sampling time point, flasks were gently mixed to ensure the homogenous suspension of cells throughout the seawater before sampling and to re-equilibrate air in the flask headspace with sterile air. The sampling timepoints for cell concentration for PLYB11 were 22:00 h, 02:00 h, 06:00 h, 10:00 h, 14:00 h and 18:00 h local time. For strain RCC1232, the sampling timepoints were 00:00 h, 04:00 h, 08:00 h, 12:00 h, 16:00 h, 20:00 h local time. The cell concentration data at 10:00 h (PLYB11) and 12:00 h (RCC1232) local time (7 of a total of 40 sampling timepoints for each experiment, selected to represent a 24 h sampling frequency) have been previously presented in Gebühr et al. (2021).

Cell concentration was determined from a 400 µL aliquot of culture (dilution factor of 26) using an automatic cell counter (CASY Model TT, OMNI Life Science). Cell concentrations are reported as viable cells measured between 3.00 to 20.03 µm using a 60 µm capillary, the lower size threshold having been determined for these strains prior to the start of the experiment. The CASY cell counter also reports particle size distribution in the sample and here we report 'cell' size from CASY derived at 4 h time intervals (coccoliths not removed from cells) from the mean particle volume measured between 3.00 to 20.03 µm on the same aliquot. Whilst we refer to CASY-derived mean particle size measurements as 'cell' size in this study, note that the particle size reported by CASY is based on the electrical resistance of particles and therefore represents an intermediate size between cell and coccosphere size (Gerecht et al., 2015), thus overestimating true 'cell' size but enabling accurate and rapid monitoring of the pace of fluctuation between smaller and larger mean particle size in the cultures for the purpose of monitoring cell division over time. For this reason, we use size measurements obtained through microscopy (Section 2.4) rather than CASY for accurate coccolith and coccosphere size data to investigate the short-term response of morphology and calcification to salinity treatments applied in this study. Particle volume measurements from CASY have a maximum error of ±2% (OMNI Life Science).

Daily growth rates $\mu_{24h}$ ($d^{-1}$) were calculated from cell concentration data following Eq. (1):

$$\mu_{24\,h} = \frac{lnN_t - lnN_{t-24h}}{t - t_{-24h}} \tag{1}$$

where $N_{t-24\,h}$ and $N_t$ are the cell concentration, N, of the culture at two sampling time points, t and $t_{-24h}$, that are consecutive but separated by 24 h, e.g. sampling at 10pm or at 2pm on two consecutive days (Guillard, 1973; Wood et al., 2005).

Instantaneous cell division rates $\mu_t$ ($h^{-1}$) were calculated for overlapping 8 h time intervals following Eq. (2) (Nelson and Brand, 1979):

$$\mu_t = \frac{1}{N_t} \frac{N_{t+4h} - N_{t-4h}}{8\,h} \tag{2}$$

## 2.4 Coccolith and coccosphere morphometrics

Samples for measurement of coccosphere diameter (size including the external layer(s) of coccoliths, $\varnothing$) and coccolith length ($C_L$) using scanning electron microscopy (SEM) were taken approx. every 8 hours beginning at 12 h. At each sampling timepoint, 2 to 5 mL of culture from each flask were gently filtered on a polycarbonate filter (0.8 µm pore size, 25 mm diameter) using a borosilicate vacuum filtration flask (Millipore) and air dried for 24 h. For each salinity treatment, one of these triplicate filters was then mounted onto an aluminium stub and sputter-coated with 4-nm platinum for SEM imaging using a field-

emission SEM (Zeiss SIGMA VP). Measurements at 0 h were taken from a single filter from the salinity 35 stock culture used to inoculate each experimental flask (i.e., the size measurements at 0 h are the same for all salinity experiments for each strain). Coccolith size measurements were made for a minimum of 50 flat-lying, individual coccoliths at a magnification of 20,000x from each filter and at least 50 intact coccospheres were imaged at a magnification of 10,000x for coccosphere size measurements. Measurements were made using ImageJ (v.1.51) from images with a dimension of 1024 x 768 pixels. Coccolith

and coccosphere size measurements were calibrated to 2 µm polystyrene calibration beads (certified mean diameter 1.998 µm ± 0.016 µm; Duke Standards Microsphere 4000 Series, certified batch number 4202–008) filtered onto a polycarbonate filter and measured once vertically and once horizontally then averaged. Coccosphere size was similarly measured as the average of one vertical and one horizontal measurement (Lamoureux and Bollmann, 2004; Nederbragt et al., 2004).

   Measurements of $\varnothing$, $C_L$ and coccolith width from a single sampling timepoint on experiment day 7 (timepoint 14.07.2014

10:00 h for PLYB11, all salinities) and at a single sampling timepoint on experiment day 5 (12.07.2014 12:00 h salinity 25 and 35) or day 7 (13.07.2014 20:00 h salinity 45) for RCC1232 have previously been presented in Gebühr et al. (2021). Morphometric data for the other approx. 18 sampling timepoints in each experiment is presented here for the first time.

## 2.5 Cellular particulate organic and inorganic carbon content

   To further quantify the impact of morphometric responses to short-term salinity stress on calcification and cellular

biogeochemical traits of *E. huxleyi*, we estimated cellular particulate inorganic carbon content (PIC), biomass (particulate organic carbon content, POC) and the inorganic to organic carbon ratio (PIC:POC) of both strains towards the end of each salinity experiment. Cellular PIC was estimated following the method of Young and Ziveri (2000), where cellular PIC is a function of $C_L$ and the number of coccoliths per coccosphere ($C_N$) following Eq. (3):

$$cellular\ PIC\ (pg\ cell^{-1}) = \left(C_L{}^3 \times 2.7 \times K_s\right) \times C_N \tag{3}$$

where $K_s$ is a shape factor that describes species-specific coccolith morphology and 2.7 is the density of calcite (pg µm$^{-3}$). Various $K_s$ values have been published for *E. huxleyi* but here we use $K_s$ values published by Linge Johnsen et al. (2019; their Table 3) derived from thickness measurements under circular polarised light microscopy that are specific to strain PLYB11 (mean $K_s$ = 0.014 for salinity 25 or 0.015 for salinity 35 and 45) and strain RCC1232 (mean $K_s$ = 0.017 for salinity 45 or 0.019 for salinity 25 and 35) and were from experiments using the same salinity conditions as applied here. Within sample 148 h

(PLYB11) and 140 h (RCC1232) from each salinity experiment, the $C_N$ of 30 individual coccospheres (the number of

coccoliths visible on each coccosphere surface multiplied by two to approximate total number of coccoliths per coccosphere) and the $C_L$ of one of the coccoliths present on the surface of the same coccospheres (i.e., distinct from the $C_L$ measured on exclusively loose, flat-lying coccoliths) were measured from each SEM image to estimate cell-specific PIC using Eq. 1. Cellular biomass (POC) was estimated as a function of cell volume. Cell diameter cannot be directly measured from SEM images (as the coccosphere obscures the internal organic biomass) and was therefore estimated for the same 30 coccospheres in 148 h and 140 h samples of both strains by subtracting 2x coccolith thickness (using mean coccolith thickness values for PLYB11 and RC1232 at salinity 25, 35 and 45 conditions from Linge Johnsen et al., 2019) from our measurements of coccosphere diameter for each cell. Cell POC was then estimated following Eq. 4 (Menden-Deuer and Lessard, 2000, for prymnesiophytes):

$$cellular\ biomass, POC\ (pg\ cell^{-1}) = 0.23(cell\ volume)^{0.9} \tag{4}$$

**2.6 Media chemistry**

The pH and total alkalinity were measured at the beginning (0 h) and end (156 h) of each salinity experiment. A portable pH meter (WTW Multi 3400i, Xylem Analytics, Germany) was used to measure pH and a titration method (MQuant Alkalinity Test, Merck) was used to measure alkalinity. Dissolved inorganic carbon (DIC) was calculated approximately as the difference between the two acid capacity values determined through titration ($K_{S4.3}$ – $K_{S8.2}$). The media chemistry at the start and end of these salinity experiments was previously reported in a supplementary dataset (Supporting information Table S4) accompanying Gebühr et al. (2021) and can be accessed through the online supporting information of that publication.

**2.7 Statistical analysis and data visualisation**

Statistical analysis was performed using GraphPad Prism for macOS (v8.4.1, GraphPad Software, LLC). A one-way ANOVA was performed to assess statistical changes in coccolith size or coccosphere diameter through time within each salinity experiment with a Tukey's post-hoc test to identify the source of the main effect determined by ANOVA. Data were considered significant at the 95% confidence interval ($p < 0.05$). Data figures were plotted in GraphPad Prism and final layout was arranged using Adobe Illustrator.

**3 Results**

**3.1 Cell division phasing and growth under three salinity conditions**

Under all salinity conditions, the mean cell size and cell concentration of *E. huxleyi* PLYB11 cultures during the experiments (note that here 'cell' size is the particle size measured using a CASY particle counter, which is an intermediate measurement between cell and coccosphere size and therefore an overestimate of true cell size; see Methods) shows a repeating cyclicity that represents a phased cell division cycle (Fig. 1a-f), i.e., a portion of (but not all) cells in the culture are moving collectively through the cell division cycle each 24 h period (see also Chisholm, 1981, for terminology). This cell division cycle phasing

persisted for the complete duration of the experiment (6.5 d) after the onset of continuous light conditions (at 0 h, previously acclimated to 12:12 L:D) under all salinity conditions. Phasing of the cell division cycle was identified by distinct maxima and minima in mean cell diameter that were repeated with an interval of 24 h ± 4 h as well as regular intervals of increasing cell concentration (indicating an interval of population division) followed by a plateau in cell concentration (indicating an interval of production, or no cell division). Intervals of increasing cell concentration aligned with the occurrence of cell size minima, as would be expected during an interval of phased cell division. Under all salinity conditions, consecutive cell size minima and periods of increasing cell concentration occurred during what would have been the last 4-8 h of the pre-experiment dark period (indicated by the alternating grey shading within the x-axis of Fig. 1).

Cyclical fluctuations in cell concentrations and cell diameter were also apparent in strain RCC1232 under all salinities (Fig. 1g-l). However, intervals of cell concentration increase/stasis and shifts from cell size minima to cell size maxima were less uniform and there was a smaller size change between successive cell diameter minima and maxima in this strain compared to PLYB11. Cell diameter fluctuations in RCC1232 were most pronounced under salinity 45 (Fig. 2k). Mean cell diameter minima occurred every 24 to 28 h under all salinity conditions, with longer intervals between cell diameter minima occurring after approx. 72 h of the experiment. Intervals of minimum cell diameter coincided with the end of an interval of increasing cell concentration, comparable to the observations for PLYB11.

The 156-h duration of the experiments represented just over six division-production cycles during which approximately one third to one half of the population moved through cell division (growth rates, $\mu_{24 h}$, ranging from 0.26 d$^{-1}$ in PLYB11 under salinity 45 to 0.41 d$^{-1}$ in RCC1232 under salinity 45). The experiment duration therefore represented between 2 and 4 complete generations under the salinity conditions. Instantaneous cell division rates ($\mu_t$, 8h averaging) fluctuated between a minimum of -0.01 h$^{-1}$ and 0.02-0.049 h$^{-1}$ at salinity 25, 0.03-0.064 h$^{-1}$ at salinity 35, and 0.015-0.033 h$^{-1}$ at salinity 45 for PLYB11 (Fig. 2a-c), with an interval of 24 h ± 4 h between peak division rates throughout the experiments. For RCC1232, instantaneous cell division rates (Fig 2e-g) fluctuated between -0.01 h$^{-1}$ (minimum for all salinity treatments) and 0.015-0.092 h$^{-1}$ at salinity 25, 0.014-0.075 h$^{-1}$ at salinity 35, and 0.017-0.044 at salinity 45 with an interval of 24 h ± 4 h between peak division rates. Division rate minima occurred during what would have been the last 4-8 h of the pre-experiment dark period or the first 4-8 h of the pre-experiment light period in both strains. In the first 48 hours, RCC1232 under salinity 25 and salinity 35 conditions shows large peak division rates that decrease by more than half by 72 h. Such a large change in peak division rates between the first 2 days and the remainder of the experiment duration is not observed in PLYB11. After 3-4 days of growth under the experimental treatments (between 80 h and 152 h), periodicity in cell division rates clearly persists (Fig. 2d, h) but peak division rates start to decline in both strains under salinity 25 and 35 by approx. 144 h in both strains (peak $\mu_t$ decreasing from 0.04-0.06 h$^{-1}$ to 0.02-0.05 h$^{-1}$ in PLYB11 and peak $\mu_t$ decreasing from 0.025-0.035 h$^{-1}$ to 0.01-0.015 h$^{-1}$ in RCC1232). In days 4-6 of the salinity 45 experiment with RCC1232, a 4-8 h offset between the timing of peak division and the timing of peak division under salinity 25 and 35 also emerges (Fig. 2h). For PLYB11, peak division rates in the final 3 days of each salinity experiment all occur at the same timepoint (Fig. 2d).

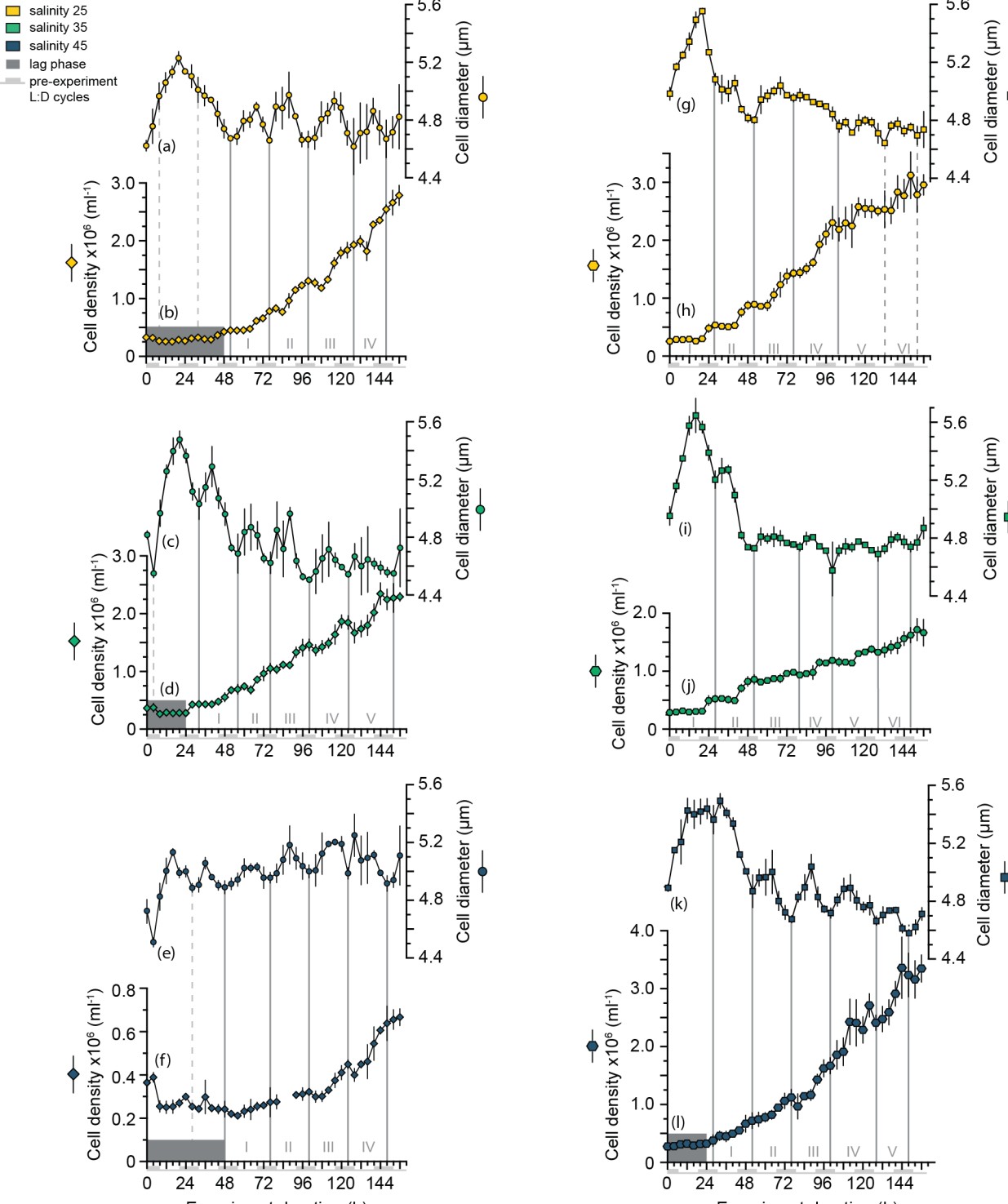

**Figure 1:** Cell density and cell size (derived from CASY particle counter) of *E. huxleyi* strains PLYB11 (**a-f**) and RCC1232 (**g-l**) over the 156-h duration of experiments under salinity 25 (hyposaline), salinity 35 (control) and salinity 45 (hypersaline) conditions. Each data point represents the mean and standard deviation of triplicate measurements. Note the different y-axis scale in (f). Sequential population division cycles are indicated by the vertical grey lines and numbered with Roman numerals, determined by mean cell size minima. Some experiments exhibit a period of minimal growth or an initial lag phase in the first 24-48 h, indicated by the dark grey shaded area. Inferred division cycles that occur during these initial lag phases are denoted by dashed grey lines. For context, the timing of pre-experiment alternations between light and dark (L:D) conditions are shown as light grey shading within the x-axis labels (note that continuous light conditions were applied for the duration of all experiments; see Methods).

Based on the logarithmic transformation of cell concentration data, all PLYB11 experiments (including control salinity 35) and RCC1232 salinity 45 experiment showed an initial lag phase of 24 h to 48 h. Mean cell size increased by 0.5-0.9 μm over the first approx. 12 h of each experiment before decreasing again over the following 12-24 h. This was equivalent to a mean cell size increase of 11% (RCC1232) or 13% (PLYB11) at salinity 25 and 45 and an increase of 14% (RCC1232) or 20% (PLYB11) at salinity 35 (Fig. 1 and 2). In RCC1232 and PLYB11 salinity 35 and 45 experiments, the subsequent mean cell size decrease was not as great as the initial size increase in the first 12-24 h of the experiment and was followed by a smaller size increase at the beginning of division cycle II (similar in magnitude to subsequent mean size minima-maxima fluctuations for the remainder of the experiment). In both strains under salinity 35, division cycle II (between 32 h and 56 h in PLYB11 and 28 h and 52 h in RCC1232) saw a large size decrease (approx. -10%) during the interval of increasing cell concentrations.

## 3.2 Effect of salinity on coccolith and cell size in PLYB11

Under control salinity 35 conditions, mean coccolith size ($C_L$) and coccosphere size ($\varnothing$) in PLYB11 remained relatively unchanged for the duration of the experiment, fluctuating between 2.6-2.9 μm and 4.6-5.2 μm, respectively, but with no consistent temporal trend (Fig. 3b and e). Following the abrupt transition to salinity 25, mean $C_L$ responded to the abrupt transition to salinity 25 with a significant decrease from 2.9 to 2.6 μm (-10%) within the first 12 h of exposure (one-way ANOVA, $F_{(7, 410)}=6.493$, $p<0.0001$; Tukey post-hoc showed that $C_L$ measured at 0 h is statistically larger than all other measured timepoints and no other multiple comparisons were statistically significant; Fig. 3a) and mean $C_L$ remained at 2.5-2.7 μm for the remainder of the salinity 25 experiment. Under salinity 45, mean $C_L$ increased in two stages (Fig 3c), with a small but significant step increase of 7% at 36 h and a further 9% increase in $C_L$ at 100 h. Mean $\varnothing$ also changed at a similar timepoint to $C_L$ change under salinity 45 (Fig. 3d-f), gradually increasing from 4.9 μm at 76 h to 5.3 μm at 100 h (an equivalent mean cell volume increase of 27%) after which mean $\varnothing$ remained relatively constant at 5.1-5.3 μm (Fig 3f). In contrast, the response of mean $\varnothing$ to salinity 25 occurred later in the experiment than the response of mean $C_L$, decreasing from 5.0 μm to 4.5 μm between 100 h and 132 h. The mean $\varnothing$ at the end of the salinity 25 experiment (156 h) was 4.2 μm (an equivalent cell volume decrease of 41%).

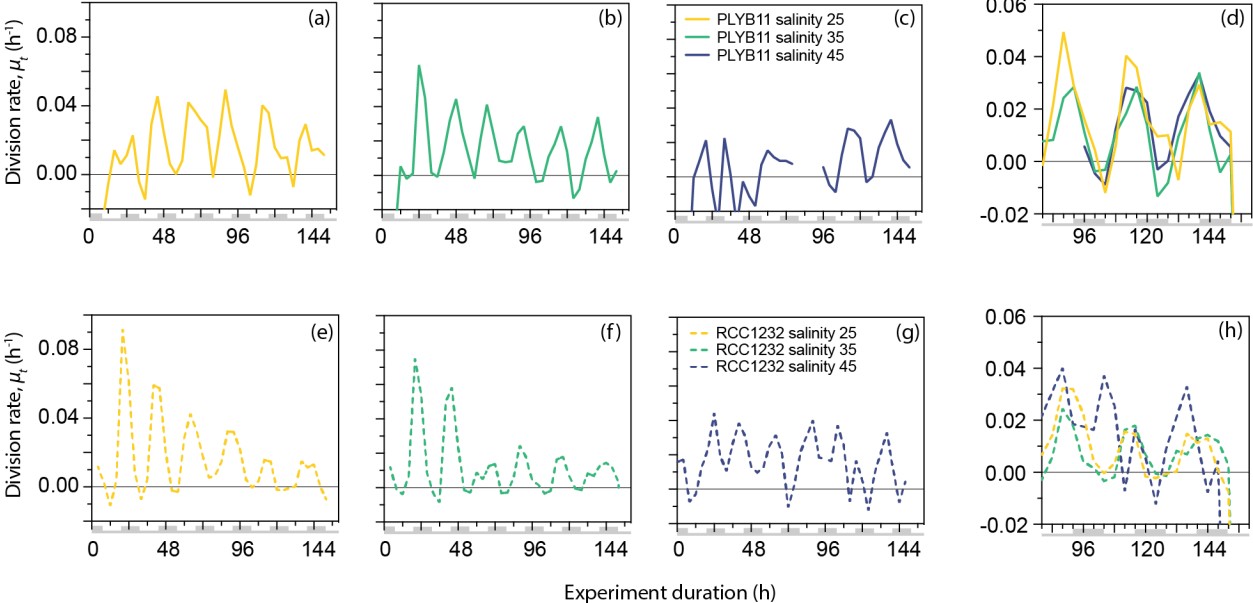

**Figure 2:** Division rates, $\mu_t$, of *E. huxleyi* strains PLYB11 **(a-d)** and RCC1232 **(e-h)** over the duration of experiments under salinity 25 (hyposaline), salinity 35 (control) and salinity 45 (hypersaline) conditions. For comparison across the three salinity treatments, division rates over days 4-6.5 of the experiments are shown in **(d)** for PLYB11 and **(h)** for RCC1232 (note the different y-axis scale). Similarly to Fig. 1,
the timing of pre-experiment alternations between light and dark conditions are shown as light grey shading within the x-axis labels for context.

As phasing of the cell division cycle persisted under continuous light for the entire duration of the experiment, the analysis of the short-term effect of salinity on coccolith and cell size must be based on samples taken at the same timepoint within the cell
cycle to avoid comparing data from later in the production phase (when cells are larger) with data from the division phase or early production phase (when cells are smaller as they have recently divided). We therefore compared temporal changes in coccolith and coccosphere size of PLYB11 under exposure to salinity 25 and 45 based on measurements taken from the same cell cycle point (cell size minima) at 76 h, 100 h, 124 h and 148 h to see if the magnitude and/or direction of salinity effects on morphology remained constant with time or changed through time as the population was exposed to the new salinity
condition for longer (Fig. 4). After just three days of growth (76 h), there is a clear difference in PLYB11 $C_L$ between low salinity, control, and high salinity conditions, with larger $C_L$ under higher salinity conditions (Fig. 4a). The size difference between salinity 35 and salinity 45 coccoliths becomes even more pronounced by days 4-6, i.e., with longer exposure to hypersaline conditions. By contrast, the effect of salinity on $\varnothing$ develops more steadily as the experiment progresses (Fig. 4b): after 3-4 days growth (76 h to 100 h), coccospheres tend to be larger at salinity 25 and 45 than at salinity 35, however after 6
310 days of growth (148 h), mean $\varnothing$ is smallest at salinity 25 and largest at salinity 45.

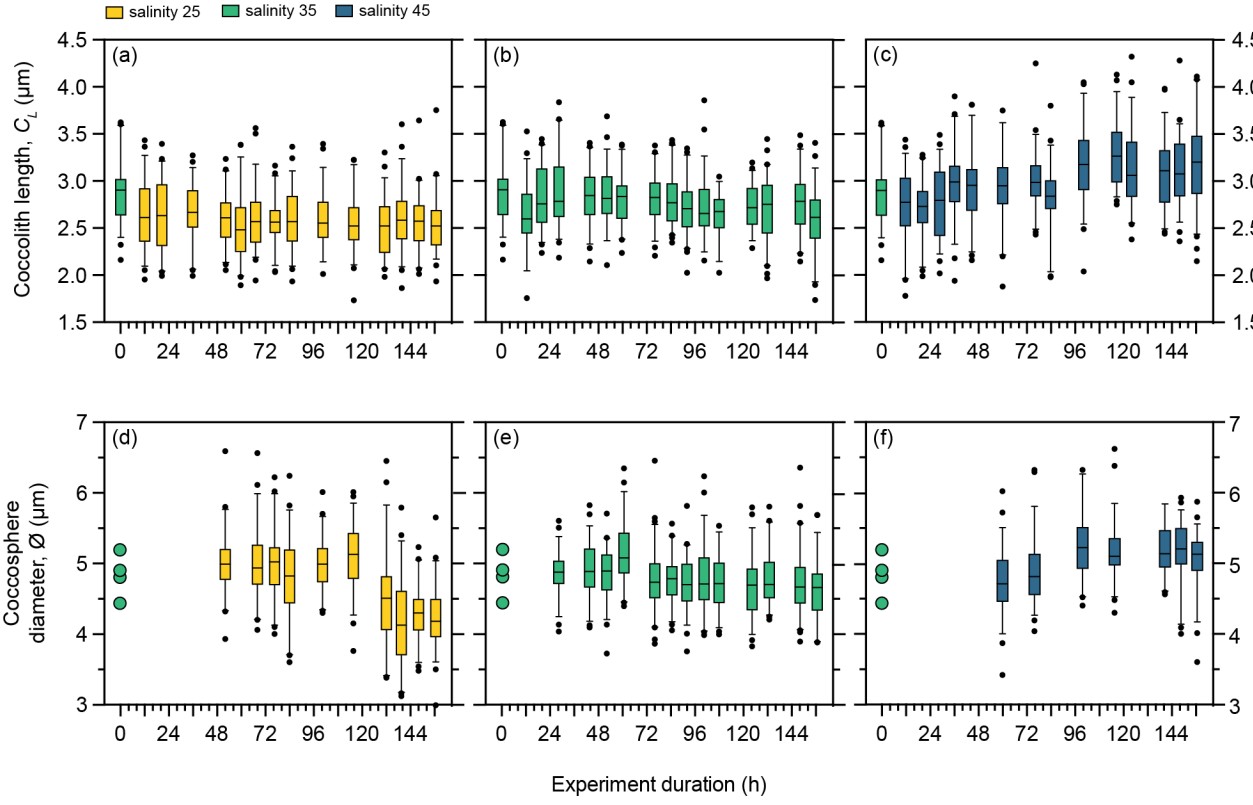

**Figure 3:** Morphology of *E. huxleyi* strain PLYB11 under three salinity conditions over 156 h exposure duration. **(a-c)** coccolith length under salinity 25, salinity 35, and salinity 45. **(d-f)** coccosphere diameter under salinity 25, salinity 35, and salinity 45. Box and whisker plots show the 25th-75th quartiles (box with median shown by the central line) and 5-95 quartiles (whiskers) of a minimum of 50 measurements at each timepoint. Measurements outside the $5^{th}$-$95^{th}$ quartiles of the data are shown as circles. Measurements at time 0 h are taken from the stock control culture (salinity 35) used to inoculate all flasks and are therefore also shown as the start coccolith length or coccosphere diameter measurements for salinity 25 and 45 experiments at 0 h. The 0 h sample for PLYB11 coccosphere diameter measurements was lost but a small number of coccosphere measurements (n=4) could be made from coccospheres imaged within the coccolith length sample and are provided (circles, d-e) to indicate coccosphere diameter at 0 h in this experiment.

### 3.3 Effect of salinity on coccolith and cell size in RCC1232

In contrast to PLYB11 under control conditions, RCC1232 grown under salinity 35 showed an increase in $C_L$ within the first 48 h (Fig. 5b; one-way ANOVA, $F_{(13, 744)}$=4.316, $p<0.0001$; Tukey post-hoc showed that $C_L$ measured at 12 h was significantly smaller than all other timepoints except 84 h, 108 h and116 h and no other multiple comparisons were statistically significant). A significant increase in $\varnothing$ within the first 24 h in the salinity 35 experiment is also clear (Fig. 5e; one-way ANOVA, $F_{(11, 647)}$=14.59, $p<0.0001$; Tukey post-hoc showed that $C_L$ measured at 0 h is statistically smaller than all other measured timepoints) and persisted for the remainder of the experiment (Fig 5b, e).

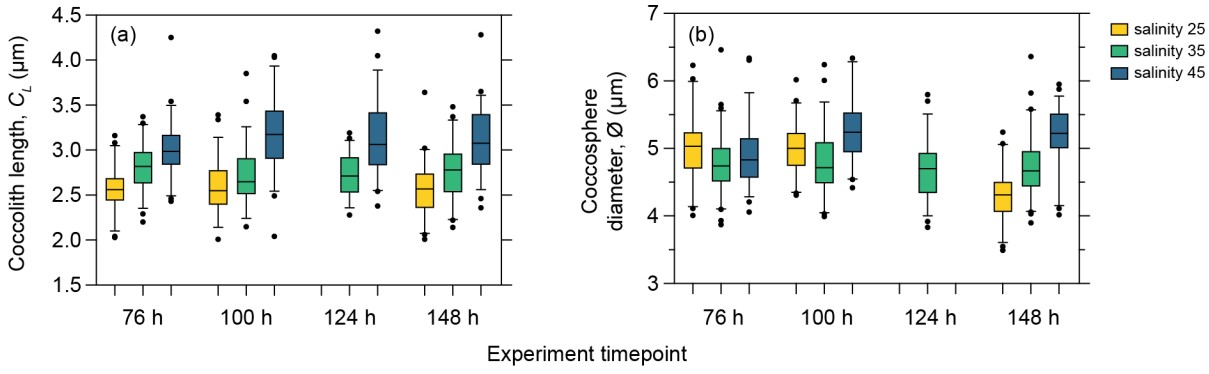

**Figure 4:** Coccolith length **(a)** and coccosphere diameter **(b)** response of *E. huxleyi* strain PLYB11 after 76 h (approx. 3 days), 100 h (approx. 4 days), 124 h (approx. 5 days) and 148 h (approx. 6 days) exposure to salinity 25, salinity 35 and salinity 45 conditions. The measurements from these selected timepoints are sampled from the same cell cycle timepoint (cell size minima). Samples for $\varnothing$ were not taken at 124 h from the salinity 25 and salinity 45 experiments.

Mediterranean strain RCC1232 showed a negligible $C_L$ response to salinity 25 relative to $C_L$ at 0 h but $C_L$ increased by approx. 18% under salinity 45 conditions within the first 36 h of the experiment (Fig. 5a,c). A significant 28% increase in $\varnothing$ is recorded between 0 h and 36 h under salinity 25 (Fig. 5d; Tukey multiple comparison test, p<0.0001, 95% C.I.=-1.550 to -0.9713). Intact coccospheres were rarely observed on filters after 72 h growth under low salinity (Fig. S1 of Gebühr et al., 2024; see Data Availability statement). RCC1232 mean $\varnothing$ increased by 27% over the course of the salinity 45 experiment but the $\varnothing$ increase primarily occurred two days earlier, within the first 48 h (Fig. 5f).

To assess the overall impact of low and high salinity treatments on $C_L$ and $\varnothing$, measurements from the same points of the cell division cycle were shown for PLYB11 in Fig. 4. A comparable analysis was not possible for RCC1232 because fewer overall SEM measurements of $\varnothing$ were taken (and were not possible under salinity 25, as explained above) and because the timepoints of $C_L$ SEM measurements unfortunately did not align well with the timepoints of cell size minima/maxima as determined from CASY data (Fig. 1g-l). However, after 3-4 days of growth under each salinity conditions, RCC1232 $C_L$ was consistently smallest under salinity 25 and largest under salinity 45 (Fig. 5). RCC1232 $\varnothing$ under each salinity was less variable across the different salinity conditions than observed in PLYB11 (based on SEM measurements; Fig. 3-5) but also appears to have not completely stabilised after >6 days under the new salinity conditions (based on CASY measurements; Fig. 1). Under all salinity conditions, size measured at 0 h from the control 35 salinity inoculum was also substantially smaller than $\varnothing$ measured at all other timepoints (Fig. 5).

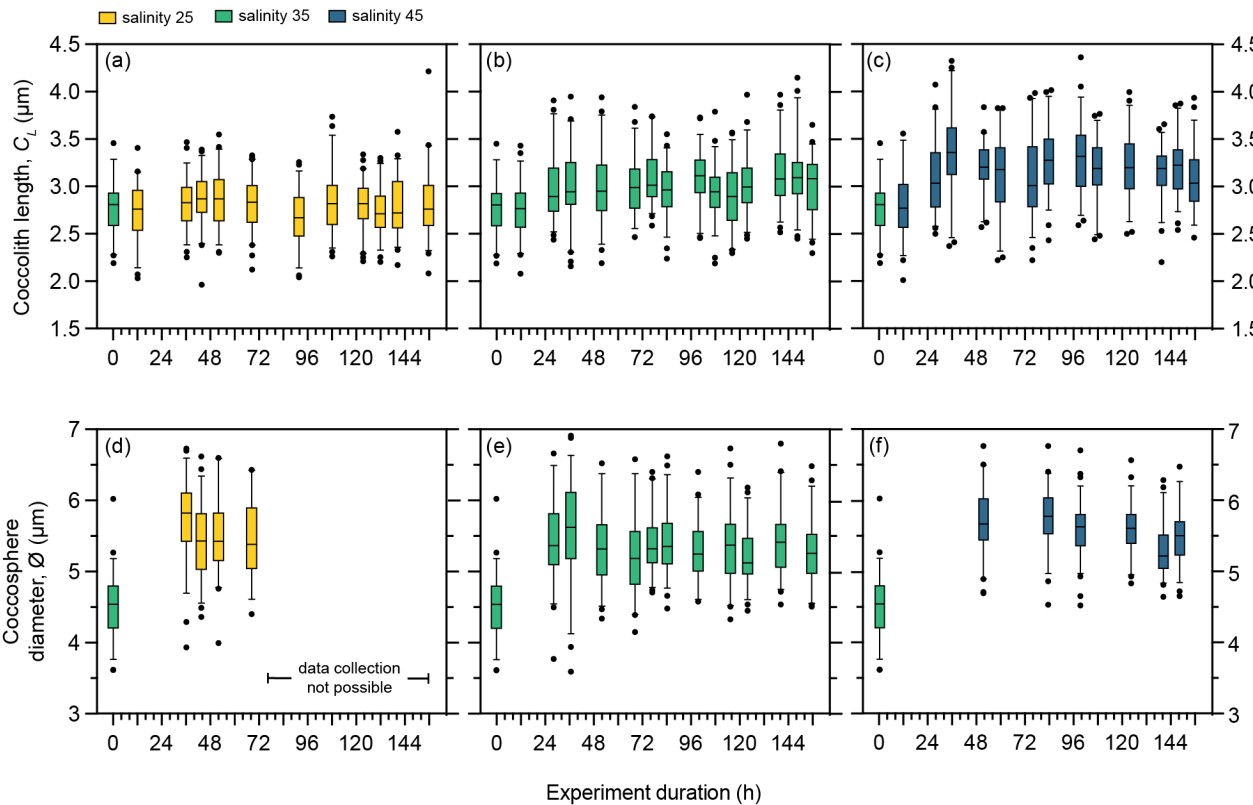

**Figure 5:** Morphology of *E. huxleyi* strain RCC1232 under three salinity conditions over 156 h exposure duration. **(a-c)** coccolith length ($C_L$) under salinity 25, salinity 35, and salinity 45. **(d-f)** coccosphere diameter ($\varnothing$) under salinity 25, salinity 35, and salinity 45. Box and whisker plots show the 25th-75th quartiles (box with median shown by the central line) and 5-95 quartiles (whiskers) of a minimum of 50 measurements at each timepoint. Measurements outside the $5^{th}$-$95^{th}$ quartiles of the data are shown as circles. Measurements at time 0 h are taken from the stock control culture (salinity 35) used to inoculate all flasks and are therefore also shown as the start $C_L$ or $\varnothing$ measurements for salinity 25 and 45 experiments at 0 h. Coccosphere measurements after 96 h were not possible for the salinity 25 experiment as severe coccolith malformation led to all coccospheres collapsing when filtered.

### 3.4 Calcification response to salinity

Our short-term experiments show that the rapid effect of salinity on *E. huxleyi* morphology leads to changes in the inorganic carbon per coccolith and per cell of both strains between each salinity condition (Table 1). After approx. 6 days growth, the difference in the size of PLYB11 coccoliths between salinity 25 and salinity 45 conditions (+22%) in PLYB11 is equivalent to a 95% increase in coccolith particulate inorganic carbon (PIC) compared to PIC under salinity 25 (Table 1). In RCC1232, mean coccolith PIC was 26% larger under salinity 45 conditions relative to salinity 25 conditions after approx. 6 days growth but coccolith PIC was conversely largest at salinity 35, as mean $C_L$ was only 0.03 μm smaller at salinity 35 than salinity 45 but the mean $K_s$ value of RCC1232 at salinity 35 compared to salinity 45 is 12% larger (Linge Johnsen et al., 2019). At the

cellular level, PIC per coccosphere after approx. 6 days growth under salinity 45 is almost double that at salinity 25 in PLYB11 (an increase of 1.38 pg C cell$^{-1}$ or 99%), principally driven by the response of $C_L$ to salinity as the number of coccoliths per cell ($C_N$) remains relatively unchanged between salinity conditions. Cellular PIC could not be estimated for RCC1232 under salinity 25 conditions as coccospheres were too poorly preserved (Fig. S1). Cellular PIC was 4% larger under salinity 45 compared to salinity 35 in RCC1232 after approx. 6 days growth. Larger coccosphere (cell) size with increased salinity

additionally translates to a 55% increase in cell biomass (particulate organic carbon, POC) in PLYB11 between salinity 25 and 45 conditions. The interaction between increased $C_L$ and $\varnothing$ with salinity and variability in $C_N$ between individual coccospheres results in smaller differences in cellular PIC:POC between salinity treatments, although PIC:POC is slightly higher (approx. 8%) under salinity 45 conditions.

**Table 1.** Mean (±sd) values for morphology and cellular biogeochemical parameters of *E. huxleyi* after approx. 6 days growth under three salinity conditions. Growth rate ($\mu_{24h}$) is calculated between experiment day 2 and day 5 (see Methods). Morphometric data are from 148 h for PLYB11 and 140 h for RCC1232 (i.e., from comparable points of the cell division cycle, approximately a minimum in mean cell size). Coccolith length ($C_L$) and coccosphere diameter ($\varnothing$) are directly measured from SEM images for a minimum of 50 individuals. See Methods for the calculation of PIC (particulate inorganic carbon) following (Young and Ziveri, 2000) and POC (particulate organic carbon) following

(Menden-Deuer and Lessard, 2000) based on morphometric measurements from 30 individual coccospheres per sample.

| | $C_L$ (µm) | $\varnothing$ (µm) | PIC (pg C) | | POC (pg C cell$^{-1}$) | PIC:POC(mol:mol) |
|---|---|---|---|---|---|---|
| | | | per coccolith | per coccosphere | | |
| **PLYB11** | | | | | | |
| Salinity 25 | 2.54 (0.30) | 4.30 (0.39) | 0.077 (0.03) | 1.40 (0.43) | 6.14 (1.79) | 0.25 (0.10) |
| Salinity 35 | 2.76 (0.31) | 4.71 (0.45) | 0.110 (0.035) | 1.95 (0.72) | 8.00 (2.64) | 0.25 (0.06) |
| Salinity 45 | 3.11 (0.36) | 5.20 (0.43) | 0.150 (0.054) | 2.78 (0.74) | 9.54 (1.93) | 0.27 (0.05) |
| **RCC1232** | | | | | | |
| Salinity 25 | 2.79 (0.31) | - | 0.138 (0.22) | - | - | - |
| Salinity 35 | 3.11 (0.32) | 5.41 (0.45) | 0.191 (0.06) | 3.28 (1.06) | 11.84 (3.18) | 0.30 (0.09) |
| Salinity 45 | 3.14 (0.29) | 5.30 (0.37) | 0.174 (0.04) | 3.40 (1.07) | 11.04 (2.28) | 0.32 (0.10) |

## 4 Discussion

### 4.1 Persistent phased cell division under continuous light

    Many phytoplankton exhibit cell division cycles that are synchronised (with the population doubling each day) or phased

(where population doubling time exceeds one day) to external light: dark fluctuations (Chisholm and Brand, 1981, and references therein). Populations entrained to the light: dark cycle typically restrict cell division to a portion of the light: dark cycle (e.g., Nelson and Brand, 1979), usually the dark phase, and production occurs during the light phase (Harding et al., 1981). Cell division in *E. huxleyi* occurs primarily during the dark phase (Bucciarelli et al., 2007; Müller et al., 2008) under a range of light: dark conditions (Paasche, 1967). This phased cell division leads to diel variability in parameters that are

commonly measured to track physiological responses to changing environmental conditions (Kottmeier et al., 2020), such as cellular biomass (POC). Cell concentrations through time (<24 h measurement frequency) will similarly show steplike increases (Kottmeier et al. 2020) when population division is phased rather than the constant exponential increase in cell concentrations expected for fully desynchronised populations (Müller et al., 2015). When sampling a phased population over a period of time, measurement timepoints therefore need to be carefully selected so that they target the same point of the cell

division cycle each 24 h period (Barcelos e Ramos et al., 2010; Kottmeier et al., 2020). This is particularly important when aiming to quantify the impact of environmental perturbations on cell physiology, as absolute increases or decreases in cellular elemental content will vary throughout the day and comparing data collected from different timepoints in the cell division cycle across experiments could change the conclusions of the experiment (Kottmeier et al., 2020).

    To circumvent this complexity, some experimental designs apply continuous 24 h light to cultures with the aim to fully

desynchronise the cell division cycle. For a fully desynchronised population growing under continuous light, the sampling time has no influence on measurement values as all measurement timepoints are representative of the daily mean production rate, cellular elemental content or other physiological measure of interest (Jochem and Meyerdierks, 1999; Shi et al., 2009; Müller et al., 2008, 2015; Kottmeier et al., 2020). Some phytoplankton species grow poorly or not at all under continuous light (Paasche, 1967; Brand and Guillard, 1981), reportedly including the coccolithophore species *Calcidiscus leptoporus* (Brand

and Guillard, 1981), the Prymnesiophyte *Isochrysis galbana*, *Chrysochromulina* sp., and the holococcolith form of *Coccolithus pelagicus* (Price et al., 1998). However, several coccolithophore species, including *E. huxleyi*, have been shown to grow well under continuous light (Brand and Guillard, 1981; Price et al., 1998). Additionally, *E. huxleyi* has reportedly been maintained in culture for prolonged periods of time (i.e., several months) under continuous light (Shi et al., 2009; Müller et al., 2017) and several publications explicitly report that the cell cycle of *E. huxleyi* became desynchronised when grown under continuous

light (Müller et al., 2008, 2017, 2015, 2012).

    We applied continuous light conditions from the onset of our experiments (an abrupt shift from a 12:12 light: dark cycle to continuous light at 0 h) and our 4 h sampling regime enabled the pattern of cell division to be monitored under control, low and high salinity conditions for the duration of each experiment. Unexpectedly, clear cell division phasing persisted under all salinity conditions in both *E. huxleyi* strains for the entire duration of the experiments (>6 days representing 2-4 generations;

Fig. 1 and 2). If the onset of continuous light had desynchronised cell division, we would expect to see $\varnothing$ stabilise around a constant value and continuously increasing cell concentrations through the course of the experiment (Müller et al. 2015). Instead, we see clear mean size minima and maxima in all experiments for both strains corresponding to steplike increases and plateaus in cell concentrations that correspond closely to the timings of the light: dark alternations that preceded the onset of the experiments (Fig. 1). Towards the end of the experiments, declining maximum division rates under salinity 25 and 45 in

PLYB11 (Fig. 2) may indicate that phased division of the population was starting to weaken in this strain under low and high salinity stress. However, it is not clear for how many more days phased division might have persisted beyond the end of the experiment. Declining division rates at the end of an experiment may also be an indicator that phased cell division continued but population growth was entering early stationary phase, i.e., growth could no longer proceed exponentially due to one or

more changes in the physiochemical conditions of the experiment as cell density increased through the duration of the experiment. Cell concentrations were dense towards the end of all experiments ($0.7$-$2.8\times10^6$ cells mL$^{-1}$ in PLYB11, $1.7$-$3.3\times10^6$ cells mL$^{-1}$ in RCC1232; Fig. 1), which resulted in a DIC increase of up to 18% in PLYB11 experiments and a DIC decrease of up to 18% in RCC1232 experiments by the end of experiments relative to starting media chemistry (Gebühr et al. 2021). The effect of cell density on media chemistry, nutrient availability, and/or light environment may have been sufficient to initiate a departure from exponential growth by the end of the 6.5 d experiment duration. This is indicated under all experimental conditions for RCC1232, where growth rates calculated from cell concentration data at 24 h intervals suggest non-exponential growth after approximately 5 days (120 h in salinity 25 and 35 experiments) or 6 days (144 h under salinity 45) of experiment duration (Gebühr et al. 2021). Mean daily growth rates indicate that PLYB11 maintained exponential growth throughout the duration of the experiments (Gebühr et al. 2021).

Persistent cell cycle phasing for three days after transitioning from a light: dark cycle regime (14:10 L:D) to continuous light has previously been shown for two species of *E. huxleyi* (with a division rate of approx. 1 d$^{-1}$) as well as the coccolithophore *Chrysotilia carterae* (previously *Hymenomonas carterae*) and three other marine phytoplankton species (Chisholm and Brand, 1981). Cell division phasing of *E. huxleyi* strain CCMP 371 was reportedly desynchronised by "…illumination over several generations with continuous light…" (Müller et al., 2008) but did not further clarify how long this took to achieve. Where publications report the use of continuous light, the pre-experiment acclimation period is stated to be between approx. 7 and 20 generations of growth (Zondervan et al., 2001, 2002; Müller et al., 2008; Bretherton et al., 2019). Assuming that these acclimation periods were sufficient to desynchronise the cell division cycle, we conclude that phasing of cell division to the pre-experiment light: dark conditions in *E. huxleyi* must persist for a minimum of 3-4 generations (as shown by Chisholm and Brand 1981 for populations dividing approximately daily and our experiments with lower growth rates) up to approx. 15 generations or longer. This would equate to 2-3 weeks of growth under experimental conditions for populations dividing with a growth rate of 0.7 d$^{-1}$ and approx. 4-5 weeks for populations dividing with a growth rate of 0.35 d$^{-1}$. Interestingly, cell cycle entrainment to light: dark cycles may be highly persistent in some strains or in combination with certain environmental stressors, as a previous publication reports culturing *E. huxleyi* for six months under continuous light to ensure complete desynchronisation of the division cycle (Müller et al. 2017).

**4.2 Growth and morphological responses to continuous light**

Many phytoplankton, including coccolithophores, have higher growth rates under higher irradiance level, longer daylength, or when continuous light is used (e.g., Chisholm and Brand, 1981; Harris et al., 2009; Sheward et al., 2023). We did not measure growth rate before the start of the experiment, so are unable to quantify the impact of the abrupt transition from a 12:12 L:D cycle to continuous light on growth rate under control conditions. However, both strains showed an initial phase of low or negligible increase in cell concentrations over the first approx. 24 h in the control experiment, suggesting that an immediate physiological response to the onset of continuous light probably occurred in all experiments. There are conflicting reports as to whether *E. huxleyi* has higher (Chisholm and Brand, 1981; Price et al., 1998; Bretherton et al., 2019), lower (Van Rijssel

and Gieskes, 2002), or comparable (Zondervan et al., 2001; Rost et al., 2002; Zondervan et al., 2002; Nielsen, 1997) growth rates under continuous light compared to a light: dark regime. The effect of continuous or discontinuous irradiance on *E. huxleyi* growth rate is likely to be strain-specific (Price et al., 1998; Bretherton et al., 2019) and/or vary depending on

combination of irradiance level and daylength used (Paasche, 1967; Rost et al., 2002). Subsequently, we cannot rule out that a growth rate response to the abrupt onset of continuous light contributed to growth rate differences between low and high salinity conditions over the course of our experiments. However, we note that Rost et al. (2002) reports that growth rates of the same strain of *E. huxleyi* (PLYB91/11) are comparable under both 16:10 L:D cycle and continuous light when grown under similar irradiance levels (70-100 µmol photons m$^{-2}$ s$^{-1}$).

In the RCC1232 control experiment, both $C_L$ and $\varnothing$ increase within approx. 28 h of inoculation and then remain at these larger mean values (with some fluctuations) for the remainder of the experiment (Fig. 5b,e). When combined with the overall $C_L$ and $\varnothing$ response to low and high salinity conditions over the course of the experiment, this unexpected increase in $C_L$ and $\varnothing$ under control conditions strongly influences the overall response of $C_L$ and $\varnothing$ across salinities 25, 35 and 45 by 156 h in this strain: $C_L$ increases from salinity 25 to salinity 45 whereas $\varnothing$ is broadly comparable across all salinity treatments as $\varnothing$ increased

similarly (Fig. 5). The initial shift in $C_L$ and $\varnothing$ is unlikely to be caused by measurements taken at different points in the cell division cycle (Fig. 1). Instead, the onset of continuous light may have driven a rapid morphological response in RCC1232 that is not observed in PLYB11. Larger cell sizes have previously been reported within 5 h after exposure to higher light conditions for a non-calcifying strain of *E. huxleyi* (Darroch et al., 2015) but in contrast, no significant cell size difference was reported between continuous light and 14:10 L:D experiments for at least one *E. huxleyi* strain (Price et al., 1998).

**4.3 Rapid morphological responses to abrupt salinity stress**

The physiology, cellular composition, and gene expression of *E. huxleyi* can rapidly respond to abrupt and short-term (hours to days) changes in carbonate chemistry, light environment, and nutrient levels (e.g., Barcelos e Ramos et al., 2010, 2012; Iglesias-Rodriguez et al., 2017; Darroch et al., 2015). Experiments with other phytoplankton groups have demonstrated that sudden salinity perturbations induce a cascade of rapid metabolic responses within cells, some of which may be coupled with

changes in cell size. For example, it only takes seconds to minutes for the green halophilic algae *Dunaliella* to adjust to osmotic differences between the external medium and the cell cytoplasm through rapid, passive water efflux or influx (Weiss and Pick, 1990), which simultaneously drives changes in cell size and volume (Maeda and Thompson, 1986). The impacts of this passive osmotic adjustment on cell size persist for minutes to hours (Weiss and Pick, 1990; Maeda and Thompson, 1986) depending on how rapidly *Dunaliella* restores its ionic equilibrium, e.g., through regulating glycol metabolism (Borowitzka, 2018). To

date, the short-term (hours to days) sequence of metabolic responses of *E. huxleyi* under salinity stress remains unknown. Whilst our 4 h sampling frequency is insufficient to capture cell size changes occurring due to turgor pressure adjustment within the first seconds to minutes of exposure to hyposaline (salinity 25) and hypersaline (salinity 45) conditions, as observed

in *Dunaliella*, our experiments do capture rapid changes in *E. huxleyi* growth and morphology over hours to days following the onset of salinity stress.

Both strain PLYB11 and RCC1232 exhibit type 'A' coccolith morphologies (Fig. S1; Young et al., 2003) but showed strain-specific differences in morphological characteristics under control conditions (see Gebühr et al., 2021 for a detailed morphological description of both strains), notably that RCC1232 coccolith and coccosphere sizes were on average 15% larger than PLYB11 coccolith and coccosphere size. The timescale of $\varnothing$ and $C_L$ response to abrupt salinity stress (relative to initial strain-specific $\varnothing$ and $C_L$; Fig. 3 and 5; see also Gebühr et al. 2021) was varied and did not show a consistent relationship to

strain, salinity treatment or morphological parameter. This is perhaps not surprising, as the response of cell/coccosphere size to salinity in *E. huxleyi* (Saruwatari et al., 2016; Xu et al., 2020b; Gebühr et al., 2021; Hermoso and Lecasble, 2018) and closely related *Gephyrocapsa* species (Hermoso and Lecasble, 2018) seems to be strain-specific (Hermoso and Lecasble, 2018; Gebühr et al., 2021). In addition to differences in the magnitude of cell/coccosphere size response to salinity 25 and 45 conditions between the two strains (Fig. 3-5; Table 1), Norwegian strain PLYB11 generally showed morphological changes

after a longer period of exposure (96-120 h) to salinity 25 and 45 conditions compared to Mediterranean strain RCC1232 (which showed $\varnothing$ changes within 28-36 h). As the concentration of cells in each experiment was high ($10^5$ to $10^6$ cells mL$^{-1}$) by the end of each experiment (Fig. 1), we cannot rule out that changes in media chemistry, nutrient availability or light environment may have additionally influenced coccolith length and/or coccosphere diameter. However, there is no systematic relationship between coccolith length and cell density, changes in coccolith and coccosphere size are already detectable within

2 days of abrupt salinity stress (when cell density was lower), and no statistically significant change in $C_L$ was observed in PLYB11 under control (salinity 35) conditions. This strongly supports an interpretation that abrupt salinity stress rather than the potential impact of cell density on experimental condition was the primary driver of morphological changes in the experiments.

As cell size responds to the regulation of cell turgor pressure in phytoplankton and other plants (e.g., Kirst, 1990) and references

therein), it is plausible that different timescales of morphological response are related to different rates of osmotic adjustment between strains, through osmolyte synthesis, active transport mechanisms, and/or membrane pumps under different salinity conditions. Species-specific synthesis of osmolytes and morphological responses to salinity stress have, for example, been reported for marine diatoms (Helliwell et al., 2021, and references therein) but are currently largely unknown for coccolithophores. *E. huxleyi* is a recognised producer of dimethylsulfoniopropionate (DMSP), a compatible solute that

contributes to cellular osmotic balance (Kirst, 1996). Cellular concentrations of DMSP are coupled to salinity in many phytoplankton (e.g., Keller and Korjeff-Bellows, 1996; Kirst, 1996; Stefels, 2000; Dickson and Kirst, 1987) and in the macroalgae *Ulva* (Van Alstyne et al. 2023). Intracellular concentrations of DMSP have been shown to correlate with salinity in one open-ocean strain of *E. huxleyi* (McParland et al., 2020), *E. huxleyi* strains from a range of environments (Fielding, 2010), and in the coccolithophore species *Gephyrocapsa oceanica* (Larsen and Beardall, 2023) and *Chrysotila carterae*

(Vairavamurthy et al., 1985). Cellular DMSP also responds rapidly to some environmental stressors, as quickly as within 4 h under elevated light (Darroch et al. 2015), and the elevated DMSP content of Prymnesiophyceae relative to other phytoplankton

groups (Keller et al., 1989; McParland and Levine, 2019) may provide an initial reserve to better tolerate rapid-onset salinity stress (Kirst 1996). However, further investigation is needed to identify the rate of change in cellular ion and osmolyte concentrations for *E. huxleyi* under a range of salinity conditions and strain-specific strategies for osmotic adjustment, as both

may impact the capacity of *E. huxleyi* to respond rapidly to the onset of salinity stress.

Changes in $C_L$ with salinity have previously been attributed to changes in the size of the coccolith vesicle proportional to cell volume increase/decreases that occur when water influx/efflux is used to maintain cell turgor pressure (Bollmann et al., 2009; Gebühr et al., 2021). Particularly within short-term experiments as applied here, this hypothesis implies four things: 1) that $\varnothing$ response must precede a $C_L$ response, 2) that the size change of $C_L$ and $\varnothing$ must be positively correlated over some reasonable

timeframe (Suchéras-Marx et al., 2022), i.e., cell size does not increase whilst coccolith size decreases or *vice versa*, 3) that the relative magnitude of the cell size change and the corresponding coccolith size change are reasonably proportional (i.e., a small cell size change does not drive a disproportionally large change in coccolith size), and 4) that a change in $\varnothing$ due to one or more mechanisms to maintain cellular homeostasis must persist for sufficient time for a corresponding increase in $C_L$ to be quantifiable (relative to coccolith production rates).

Overall, our data are consistent with these criteria (Fig. 3 and 5). However, we do see intervals within each experiment where large steplike changes in $\varnothing$ do not correspond proportionally to changes in $C_L$ over the following hours to days (e.g. both strains under salinity 25; Fig. 3 and 5). In some cases (e.g., both strains under salinity 45) we also see $\varnothing$ and $C_L$ changes that occur relatively synchronously (although there is some discrepancy in the $C_L$ and $\varnothing$ sampling timepoints). Generally, changes in $C_L$ tend to emerge more gradually and after a longer period of exposure than changes in $\varnothing$ (Fig. 3 and 5), apart from

RCC1232 under salinity 45, where the $C_L$ increases between 24 and 36 h occur within the same timeframe as the $\varnothing$ increase. The measured loose coccoliths initially represent a mix between coccoliths produced during pre-experiment conditions and coccoliths produced under salinity stress, with a diminishing contribution from pre-experiment coccoliths to $C_L$ measurements as the experiment progresses. New coccoliths are produced at least every approx. 60 minutes (Paasche, 2002; Suchéras-Marx et al., 2022) and cells must produce approx. 6-10 new coccoliths between each cell division to ensure a complete cell covering

for two daughter cells (based on mean $C_N$ 18-20 in our samples). It would therefore take more than one generation of growth (>24 h when $\mu<0.7$ d$^{-1}$) under new environmental conditions before $C_L$ measurements start to reflect the increasing proportion of coccoliths of a different size produced in response to physiological adjustments to the new environment (e.g., salinity). However, coccoliths are produced quickly enough that coccolith size responses should be already evident after two generations, supporting the timescale of morphological responses observed in our 156-h experiment. This might also explain why Iglesias-

Rodriguez et al. (2017) did not observed any difference in $C_L$ over a 72-h exposure of *E. huxleyi* strain NZEH to two different low pH deep seawater conditions even though coccosphere volume was significantly different (+30%) between the two deep seawater conditions after 72 h.

## 4.4 Intraspecific variability in *E. huxleyi* coccolith PIC – methodological considerations

Our estimates of coccolith PIC for strains PLYB11 (ranging from 0.04 to 0.38 pg C coccolith$^{-1}$) and RCC1232 (ranging from 0.06 to 0.39 pg C coccolith$^{-1}$) are comparable to the range of values reported for *E. huxleyi* morphotype A in the subantarctic zone of New Zealand by Saavedra-Pellitero et al. (2023) (0.02-0.35 pg C) and *E. huxleyi* in South Atlantic surface sediment samples (0.02-0.58 pg C; Horigome et al., 2014). However, our mean coccolith PIC values under each salinity treatment (Table 1) are on the lower end of the range of coccolith PIC values reported for *E. huxleyi* morphotype A coccoliths on the Patagonia shelf by Poulton et al. (2011) (0.12-0.42 pg C) and from morphotype A strains in culture (0.12-0.36 pg C under a range of temperatures; Rosas-Navarro et al., 2016, and 0.49-0.94 pg C in nutrient replete cultures; Müller et al., 2012).

To assist with the comparison of our dataset and other published coccolith PIC estimates for *E. huxleyi*, we briefly address some of the factors that may contribute to species-specific variability in coccolith and coccosphere PIC estimates between publications. Firstly, we have used a well-established morphometric-based approach to estimate coccolith and coccosphere calcite (Young and Ziveri, 2000), which is a function of coccolith size and a species-specific shape factor. Any difference in coccolith size between cultures (e.g., due to strain-specific morphological differences or coccolith size responses to environmental treatments) therefore influences morphometric-based coccolith PIC estimates. For example, Southern Ocean *E. huxleyi* morphotype A isolates have larger mean coccolith size (3.19-3.48 µm; Valença et al., 2024) compared to the mean coccolith size of the Norwegian and Mediterranean strains used here (2.54-3.24 µm; Table 1), which contributes to the higher morphometric-based estimates of mean coccolith PIC for Southern Ocean strains (0.22-0.28 pg C; Valença et al., 2024) compared to PLYB11 (0.11 pg C) and RCC1232 (0.19 pg C). Here, we have also used previously determined strain- and salinity-condition-specific shape factors in our morphometric-based coccolith PIC estimates (Linge Johnsen et al., 2019) rather than a general *E. huxleyi* shape factor (for instance, $K_s$ = 0.02 is recommended by Young and Ziveri, 2000). However, shape factors can vary within a species by up to 20% (Young and Ziveri, 2000) and coccolith calcite mass can be decoupled from changes in coccolith size (Linge Johnsen and Bollmann, 2020a), factors that can introduce uncertainty in morphometric-based coccolith PIC estimates. Coccolith PIC estimates from natural populations will also span a greater range of genetic and phenotypic variability (and therefore coccolith size) than measured in strain-specific laboratory cultures, which will also introduce variability in mean coccolith PIC for the same morphotype.

Morphometric-based estimates of coccolith PIC are generally in good agreement with estimates derived from other commonly applied methods, including birefringence-based thickness estimates of coccolith mass (Linge Johnsen and Bollmann, 2020b; Valença et al., 2024). Whilst a detailed discussion of alternative methods for estimating coccolith PIC are beyond the scope of our study, for the purpose of contrasting our estimated coccolith PIC values with published data derived from alternative methodologies, we note that coccolith PIC values derived from birefringence-based approaches have limitations related principally to coccolith crystallography and the choice of microscopy settings and calibration methods (e.g., Bollmann, 2014; Beaufort et al., 2014, 2021; González-Lemos et al., 2018; Linge Johnsen and Bollmann, 2020b). For instance, estimates of coccolith PIC derived from the birefringence-based method first published by Beaufort (2005) require a correction of the

coccolith length because of the limited detection of the outer edge of the distal shield of *E. huxleyi* coccoliths when observed under cross-polarised light microscopy (e.g., Beaufort et al., 2008). However, the correction of the underestimated length by a factor of 1.25 (Beaufort et al., 2008; D'Amario et al., 2018) may subsequently lead to an overestimation of coccolith calcite content as the undetected mass of the coccolith is estimated based on the birefringence of the thickest part of the coccolith. For

further details, refer to Bollmann (2014) and Linge Johnsen and Bollmann (2020b). It has previously been noted that estimates of *E. huxleyi* coccolith calcite from birefringence-based methods can be higher than morphometric-based estimates of coccolith PIC made on the same morphotype and/or in similar regions (Poulton et al., 2011; Saavedra-Pellitero et al., 2023). Such methodological considerations contribute to intraspecific variability in coccolith PIC reported across different studies in addition to the effect of any environmental treatment on morphology.

**4.5 Implications of rapid responses to salinity stress**

Whilst our morphological results represent changes over just 2-4 generations of growth under salinity stress, a general pattern of smaller cells with smaller coccoliths under hyposaline conditions (reduced cellular PIC with comparable cellular PIC:POC) and larger cells with larger coccoliths under hypersaline conditions (increased cellular PIC with comparable cellular PIC:POC) emerges from both strains investigated here (Fig. 6; Table 1). A similar pattern of smaller coccoliths under lower salinity and

larger coccoliths under higher salinity was also found in acclimated cultures of several different *E. huxleyi* strains, including some open ocean strains (Green et al., 1998; Paasche et al., 1996; Fielding et al., 2009; Linge Johnsen et al., 2019; Saruwatari et al., 2016), in plankton samples (Bollmann et al., 2009), and in sediment core-top samples (Bollmann and Herrle, 2007). The response of coccolith morphology and coccosphere (cell) size to the abrupt change in salinity conditions equates to a rapid (within hours to days) calcification response of *E. huxleyi* to salinity stress, with lower cellular PIC under hyposaline

conditions, increased cellular PIC under hypersaline conditions, and similar cellular PIC:POC across all salinity conditions due to the comparable timing of cell (coccosphere) size responses to salinity stress (Table 1). Our results show that *E. huxleyi* morphology and calcification is therefore sensitive to even relatively short-term (days to weeks) intervals of abrupt salinity change and, based on evidence from the literature, that these morphological effects are sustained when the new salinity condition persists for weeks to months (as would be typical for experiments using acclimated cultures).

Changes in coccolith size and coccolith morphology more generally are widely used as proxies for past environmental conditions through time (e.g., Bollmann, 1997; Henderiks and Bollmann, 2004), including paleosalinity (Bollmann et al., 2009; Herrle et al., 2018; Bollmann and Herrle, 2007). The response of *E. huxleyi* coccolith size to salinity stress within hours of exposure in our experiments signifies that transient and extreme salinity events will affect coccolith size in plankton samples alongside longer-term seasonal to decadal salinity trends. Seasonally, calcification responses on such short timeframes may

influence surface ocean alkalinity and inorganic carbon export in regions where *E. huxleyi* is a large component of the phytoplankton community. The analysis of coccolith size through time from marine sedimentary records remains most suited to capturing short- to longer-term salinity fluctuations, as sedimentation rates in laminated sediments capture seasonal to annual timescales up to sedimentation rates of thousands to hundreds of thousands of years per cm sediment in deep ocean sediments.

However, we emphasise that the rapid morphological response of *E. huxleyi* to salinity (shown here) and other environmental variables, including $CO_2$ concentrations (Barcelos e Ramos et al., 2010) and exposure to distinct water masses (Iglesias-Rodriguez et al., 2017), already influence morphology and calcification at a cellular level within only one or two generations of growth. Relative to the timescale of interest, variability in the magnitude and timing of *E. huxleyi* calcification responses to salinity (and likely other environmental parameters) will therefore contribute to the natural within-species variability in morphology and biogeochemical traits observed in natural *E. huxleyi* populations.

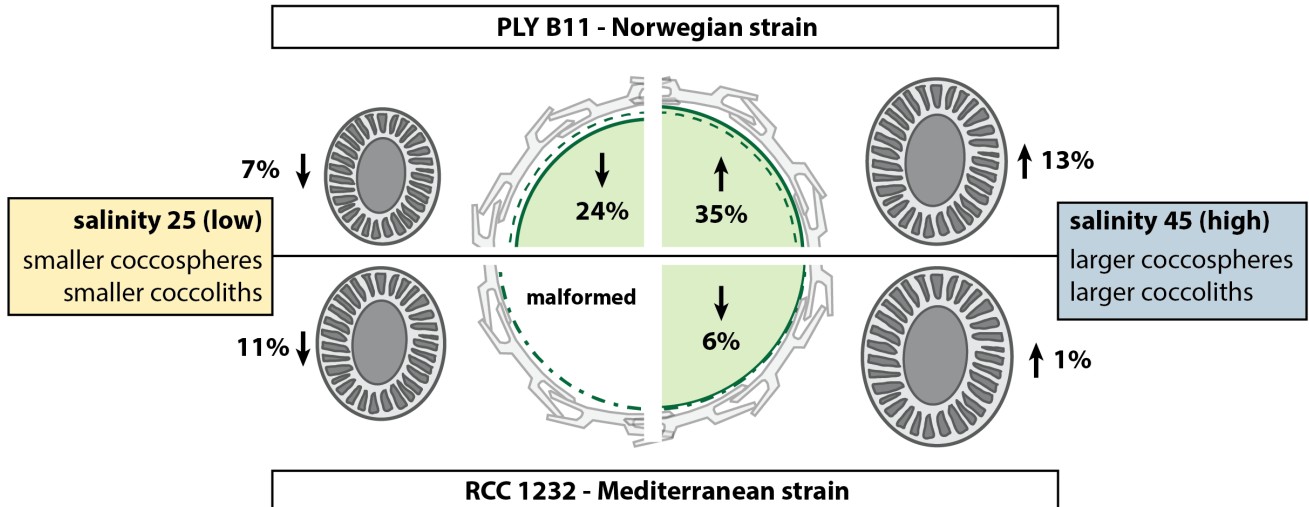

**Figure 6:** Summary of the morphological response of *E. huxleyi* strains PLYB11 (upper panel) and RCC1232 (lower panel) to abrupt exposure to low salinity 25 conditions (left) and high salinity 45 conditions (right) relative to coccosphere volume (centre cell illustration) and coccolith size (coccolith illustrations) under control salinity 35 conditions. Percentage change values are calculated based on mean $C_L$ and mean $\varnothing$ measurements at 148 h for PLYB11 and 140 h for RCC1232, which represents approx. 6 days growth under the salinity condition (see also Table 1). Dashed green line within the cell illustration indicates equivalent cell size of mean cell volume under salinity 35 conditions at 148 h or 140 h for PLYB11 and RCC1232, respectively.

**Conclusions**

The coccolithophore *E. huxleyi* has a naturally broad salinity tolerance, thriving in both relatively stable open ocean settings and the more variable environmental conditions of shelf-seas and coastal regions. Despite this salinity tolerance, the physiology and morphology of *E. huxleyi* is responsive to changes in salinity. Our experiments show, for the first time, that measurable differences in *E. huxleyi* coccolith size and coccosphere size occur within hours of abrupt exposure to hypo- and hypersaline conditions. The resultant impact of these rapid morphological responses for cellular calcification on short timescales may

impact surface ocean carbonate chemistry in regions where *E. huxleyi* is a dominant constituent of phytoplankton communities and contributes to the natural morphological variability of *E. huxleyi* coccoliths in the sedimentary record. The magnitude and timing of the response of *E. huxleyi* to salinity stress is strain-specific and may be related to different osmoregulation capacities between strains, though further exploration of the physiological and biochemical mechanisms underpinning our results was

beyond the scope of this study. Further insights would be gained from investigating the magnitude and timing of short-term morphological responses of a broader range of open-ocean and coastal *E. huxleyi* strains to a range of moderate to extreme salinity stress whilst measuring additional physiological indicators (e.g., photophysiology, carbon fixation rates, coccolith geochemistry, and cellular concentrations of DMSP and other solutes). Similarly, it is unclear whether other coccolithophore species respond similarly to the onset of salinity stress, especially as the biogeography of many species is largely restricted to

open-ocean settings with a smaller natural salinity range. If trends between morphology and salinity conditions are identified for modern representatives of longer-lived coccolithophore genera, the geological periods over which coccolith morphology can be used as an independent paleosalinity proxy could potentially be extended.

**Data availability**

Data reported in this study and supplement Fig. S1 (SEM images of coccospheres) are available at zenodo as Gebühr et al.
(2024), doi: 10.5281/zenodo.11186906. The dataset of carbonate system parameters at the start and end of experiments is available in the Supporting Information files (S4 Table) of Gebühr et al. (2021), doi: 10.1371/journal.pone.0246745.s006.

**Author contribution**

The study was conceived by JOH, CG, and JB. CG planned and conducted the experiments and data collection. Morphometric data collection was performed by CG and RS. Data analysis and interpretation was performed by RS. The manuscript was
written by RS with contributions from all authors.

**Competing interests**

The authors declare that they have no conflict of interest.

**Acknowledgments**

The authors thank the Roscoff Culture Collection at the Station Biologique in Roscoff, France, and the Plymouth Culture
Collection maintained by the Marine Biological Association in Plymouth, UK, for providing the algal strains used in this study. We are grateful to Laboratory Technician Bärbel Schminke for her assistance with culture sampling and scanning electron microscopy. This research was financed by grants awarded to JOH by the State of Hesse (Germany) in the framework of the

Initiative for the Development of Scientific and Economic Excellence (LOEWE) the Biodiversity and Climate Centre Frankfurt (BIK-F) grant number R2107 Project PB A, the DAAD-Programm Strategische Partnerschaften of the Goethe-University Frankfurt (grant number UoT6), and the Freunde und Förderer der Goethe-Universität Frankfurt am Main (funding awarded to JOH in 2010). RMS is funded by the Deutsche Forschungsgemeinschaft (DFG, German Research Foundation) project 447581699 and RMS acknowledges additional financial support through the VeWA consortium (*Past Warm Periods as Natural Analogues of our high-CO₂ Climate Future*) by the LOEWE programme of the Hessen Ministry of Higher Education, Research and the Arts, Germany.

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
