# Peer review of "Short-term response of *Emiliania huxleyi* growth and morphology to abrupt salinity stress"

_EGUsphere, 2024_

## Author Response (AR1)

**Response to reviewers comments, highlighting implemented changes in the revised manuscript**

Original reviewer comments are shown in black, our original response to reviewers comments are shown in blue and our implemented responses in the revised manuscript is shown in green. Line numbers of implemented changes refer to line numbers in the marked-up revised manuscript version that shows the changes we have made (also in green text).

**Reviewer 1:**

The study by Sheward et al. presents interesting and important results from laboratory experiments with the model coccolithophore species *Emiliania huxleyi*. In general, the study is a good fit for Biogeosciences and I acknowledge the importance of the results to improve our understanding on the ecophysiological behavior of coccolithophores in a changing environment. Two ecological different strains were submitted to abrupt changes in seawater salinity (hypo- and hypersaline) while monitoring their physiological response in terms of growth rate, cell and coccolith geometry. The response differed between the two tested ecotypes and the results give important insights on the short-term acclimation responses of coccolithophores from diverging ecological regions. The experiments were conducted using state of the art methods and the results are presented in an appropriate manner, justifying publication in Biogeosciences. However, I have some general and specific comments that the authors might want to consider to improve the quality of the manuscript.

**General**
1. The use of adverbs (e.g., also) throughout the text. I think that in many instances they could be omitted.
2. I recommend a through roughly revision of the English language and structure to improve the readability and the consistency of using scientific terms. For example, statements like "are more typically defined", "are broadly tolerant", "has largely been investigated", "slightly higher", etc., are unprecise/unscientific and should be avoided.

1 & 2: We will take on board feedback on the language use for the revised manuscript and will omit unnecessary adverbs from the text and improve statements that are generally vague. For instance, quantifying a magnitude of change rather than using general descriptive term, especially when these occur in the results or when talking about the results. We hope that this improves the readability and accuracy of the manuscript.

**Revisions made:** We have taken on board feedback on the language used through the manuscript and have omitted unnecessary adverbs from the text or exchanged them for more precise language. We have particularly focused on this in the Results section of the manuscript. For instance, we have adding statistical descriptors of magnitude of change rather than or in addition to relative descriptiors such as "slightly higher".

3. In the introduction it would be nice to state the approximate changes in salinity that are projected to be caused by climate change processes, and how these compare to modern day salinity fluctuations of coastal areas.

The approximate changes in salinity that are projected to be caused by climate change will be added to the introduction for comparison with recent observations.

**Revisions made:** We have quantified the magnitude of observed open-ocean salinity change (lines 42-43) and added the approximate changes in ocean salinity projected by climate models (lines 44-46) for context in the Introduction.

4.  Population cell densities were relatively high for experiments with *E. huxleyi* ($2x10^5$ to $3x10^6$ cells/ml). This can induce significant changes in seawater carbonate chemistry (e.g., $CO_2$ availability and pH). Additionally, it might be possible that nutrient limitation (N or P) has been induced at the end of the experimental incubations when populations reached densities above $2x10^6$ cells/ml. The authors state that pH and total alkalinity were monitored at the start and the end of the incubations. However, the data has not been presented in the manuscript. I would like to authors to show/state/discuss that changes in carbonate chemistry and nutrient availability did not significantly influence the experimental outcomes.

The carbonate chemistry data collected during these experiments has previously been reported in an earlier publication (Gebühr et al. 2021) and can be found as a supplement to this publication. As this information was not highlighted in the original submission, the availability of this information will be added to the relevant section of the Methods and the data availability section. We agree that cell density is relatively high, with the potential to affect carbonate chemistry and/or nutrient availability. The discussion will be revised in appropriate sections to state the change in carbonate chemistry between the beginning and end of the experiments and to additionally discuss the results in view of these data, e.g. adding to lines 400-403 to highlight the possibility of nutrient depleted conditions towards the end of the experiment linked to our data for division rates and discussing our coccolith size data in relation to carbonate chemistry.

**Revisions made:** The revised manuscript now references the availability of the pH and alkalinity data collected during the experiments in the Methods (lines 206-207) and in the Data Availability Statement. In the Discussion, we now state the cell concentrations and associated change in DIC between the beginning and end of the experiments (lines 433-435) and discuss the potential impact of cell density on growth (lines 435-441) and morphometrics (lines 509-515).

5.  Cell size is reported from using a CASY cell counter. However, I assume that the coccosphere size was measured. As you also report coccosphere diameter from electron microscopy analysis, it becomes confusing and the reader might think that you differentiated between cell and coccosphere sizes.

The samples run through the CASY cell counter were not acidified in advance to remove the coccosphere for the purposes of this experiment. However, CASY cell counters actually measure a size that is intermediate between coccosphere size and cell size and therefore underestimate coccosphere volume (Gerecht et al. 2015; 2018) and do not an accurate measure of cell or coccosphere size. CASY size measurements are, however, very useful for rapidly monitoring changes in size over time. We therefore agree that it is confusing to label the CASY-derived data with either "cell size" or "coccosphere size". We will instead amend reference to CASY-derived size measurements as "size" and add a statement to the methods to state that CASY measures size between cell and coccosphere size and is used to monitor the development of the cell division cycle.

**Revisions made:** To clarify this point and the difference between size measurements from CASY and from microscopy, we have added a description of the CASY measurements to the Methods (lines 144-151) and reiterated the definition of CASY 'cell' size in the Results when this CASY data is first

presented (lines 218-219) and stated that the cell size in Fig. 1 is derived from CASY in the caption to Fig.1.

6. It seems to me that the cellular and coccolith PIC content is underestimated compared to literature values (e.g., Valença et al., 2023; Müller et al, 2012; Jin & Liu, 2023), which is also reflected in the low PIC:POC ratio. Please verify and discuss this.

We have used a morphometric-based approach to estimate coccolith and cellular PIC (Young and Ziveri, 2000), which does introduce some uncertainty in the estimated PIC values. However, we aim to minimise some of the uncertainty by using shape factors that are derived from these two specific strains under the same range of salinity conditions (Linge Johnson et al. 2019). We thank the reviewer for noting that our estimated coccolith and cell PIC values may be lower than published in other literature, as this raises some additional interesting discussion points, and we confirm that it is the case that our coccolith PIC values are lower than other published values from "morphotype A" strains in some instances. For some of the examples mentioned by the reviewer, in Valença et al. (2024) morphometric-based coccolith PIC values of 0.22-0.28 pg C are reported for E. huxleyi Type A and in Müller et al. (2012), coccolith PIC for nutrient-limited E. huxleyi cultures was 0.17-0.20 pg C and for nutrient-replete cultures was 0.49-0.94 pg C (all converted from published pg CaCO3). These published values are higher than our mean coccolith PIC values for both strains across all salinity treatments (0.077-0.191 pg C), although our data are in reasonable agreement to those reported for nutrient-deplete cultures from Müller et al. (2012). There are several potential explanations for discrepancies in coccolith PIC and cell PIC between studies. Firstly, as we mention in the discussion (section 4.4), differences in coccolith size strongly influence coccolith PIC (and therefore cell PIC), especially as estimated using morphometrics. Therefore, any difference in the size of coccoliths between our cultures and those of other studies may influence the reported coccolith and cell PIC values. This also means that differences in coccolith size between strains (either in response to or independent of any environmental treatment applied) will also influence coccolith PIC. In the example of Valença et al. (2024), the Southern Ocean strains investigated had larger mean coccolith (distal shield) length (approx. 3.25 to 3.5 µm) compared to the mean coccolith length of the strains used in our experiments (2.54 to 3.14 µm). Secondly, the choice of shape factor influences morphometric-based estimates of coccolith and cell PIC. Here we use strain-specific and salinity-treatment-specific shapes factors, which may not be the case in other studies and could introduce some discrepancies, especially as the recommended shape factor for E. huxleyi stated by Young and Ziveri (2000) is greater (0.02 for morphotype A) than the values that we have used derived specifically for strains PLYB11 (0.014-0.015) and RCC1232 (0.017-0.019), which are also classified as morphotype A. Thirdly, our estimates of cell PIC are truly cellular in that they are calculated using the number of coccoliths on each individual coccosphere, which is variable within a culture. As assay-based quantification of cell PIC averages all the PIC in the culture across the number of cells filtered, it is more likely to overestimate cell PIC by also quantifying PIC in loose coccoliths and/or dead cells. These factors may contribute to differences between our estimated coccolith and cell PIC compared to other studies and between-publication differences in estimated PIC more generally. We will add to the revised manuscript a comparison between our data and those of previously published studies and discuss the possible sources of some of these differences, as described above.

Revisions made: To address this comment, we have added a new section to the Discussion ("4.4 Intraspecific variability in *E. huxleyi* coccolith PIC – methodological considerations", lines 561-602). Here, we outline how variability in the parameters used in the estimation of coccolith PIC through morphometric-based approaches can contribute to different coccolith PIC estimates between strains,

experimental conditions and studies, as well as briefly contrasting our results with other published coccolith PIC estimates from morphometric-based and birefringence-based approaches.

7. I very much appreciate the discussion regarding the transition from a phased to desynchronized *E. huxleyi* population. Indeed, from my own experimental experience, it requires an extended time. Unfortunately, I did not continuously monitor this transition and only verified the desynchronization before starting my experiments (depending on the strain and culture conditions this could involve several months). Indeed, resolving this question might be an interesting research project for future researchers and I assume that different ecotypes could be tested. I hope that the research group will have the capacities to follow this up in the near future.

**Revisions made:** No specific revisions were required for this comment, but we thank the reviewer for this positive feedback on the importance of this area of research.

**Specific comments**
1. Change "ca." to "approx.".

**Revisions made:** We have changed instance of "ca." to "approx.." throughout the manuscript.

2. Line 74: I would like to challenge the use of the term "exoskeleton" for the coccosphere and leave it to the author to decide. An exoskeleton is defined to support the body shape and to give stability for the organism. Coccolithophores, and especially *E. huxleyi*, are known to appear in an uncalcified form (naked) and do not rely on an exoskeleton for stability (or to support the cell shape). Additionally, many experiments demonstrate that removing the coccosphere (e.g., by HCl addition) does not impair growth and cellular stability. This is in strong contrast to removing an exoskeleton of, for example, an insect. Furthermore, the use of the term exoskeleton implies a protective function of the coccosphere, which is debated in the literature (e.g., Müller, 2019).

The reviewer raises interesting points about the use of "exoskeleton" to describe the coccosphere as relates to exoskeletons in other organisms and a primarily protective function. Although there is no consistently used or accepted "plain language" description for the coccosphere, we agree with the points raised by the reviewer and will reword the two instances that "exoskeleton" appears in the manuscript to remove "exoskeleton" from the description of the coccosphere.

**Revisions made:** We have replaced the two instances of "exoskeleton" in the manuscript with "cell covering" (line 17) and "an inorganic calcite cell covering (coccosphere)" (line 79-80).

3. Cellular calcite and cellular biomass are abbreviated as PIC and POC, respectively. However, these abbreviations refer to as particulate inorganic and organic carbon, respectively. This needs to be corrected throughout the manuscript as the weight of carbon vs calcite is different. I recommend to be consistent and either use cellular calcite or cellular PIC content.

We agree that the definition of calcite and particulate inorganic carbon are not interchangeable and that by implying that they are by using several terms in the manuscript, this may confuse or mislead readers. We will therefore ensure consistency through the revised manuscript and use coccolith PIC or cellular PIC exclusively.

**Revisions made:** Throughout the revised manuscript we now consistently use coccolith or coccosphere/cellular PIC when referring to our data and use calcite rarely when we are discussing more general concepts (e.g. in the Introduction).

4. Line 322: Fig. 2 does not show diameter measurements. Please verify.

**Revisions made:** This has been corrected to "Fig. 1" in the revised manuscript.

5. Citation of "Barcelos e Ramos" should be with a lower case "e" (e.g., line 373).

**Revisions made:** This has been corrected in the revised manuscript.

6. Line 586: Change "Hessen" to "Hesse".

**Revisions made:** We have not corrected this in the revised manuscript as the English pages of the LOEWE programme state "Hessen Ministry of Higher Education, Research and the Arts", so we believe that this is correct as it was originally stated (https://wissenschaft.hessen.de/ansprechen/english-information/about-the-ministry).

**Reviewer 2:**

Dr Rosie M. Sheward and colleagues evaluate the physiological and morphological response of two strains of the model coccolithophore species *Emiliania huxleyi* growing under different salinity conditions over a period of approximately one week. The selected strains were collected from the Arctic Seas (Bergen, Norway) and the temperate NW Mediterranean. The results of this study demonstrate that the physiology and morphology of *E. huxleyi* is responsive to changes in salinity with a general trend towards smaller coccospheres and coccoliths under hyposaline conditions for both strains. In turn, the response to hypersaline conditions differed across strains, with the Mediterranean strain exhibiting negligible response in the physiology or morphology while the Arctic strain produced larger coccospheres and coccoliths.
A large body of evidence indicates that coccolithophores are sensitive to projected changes in oceanic conditions driven by ongoing environmental change, such as ocean acidification, warming and nutrient availability, among others. In particular, the effect of salinity in coccolith morphometrics is largely based on field observations where a clear correlation between the morphology of E. huxleyi coccoliths in plankton and sea surface sediments and sea surface salinity variability has been documented (e.g. Bollmann and Herrle, 2007; Bollmann et al., 2009). However, there is little direct evidence from cultures about the impact of salinity changes in the physiology and/or morphology of E. huxleyi. Therefore, there is an urgent need to validate field observations through culture experiments such the one presented here.

Overall, the manuscript is clearly written, the methodology is thoroughly explained, and the results are well discussed. Moreover, figures are appropriate and of high quality. Therefore, I recommend acceptance of this manuscript after some minor/moderate corrections have been implemented:

- Authors report the use of two different strains: PLYB11 from the coastline near Bergen (Norway) and RCC1232 from the NW Mediterranean Sea. Is there a particular reason why these strains were selected? I assume authors selected strains from different environments (Arctic and temperate regions). If so, I would suggest to clearly state it in the text.

The two strains were selected from a range of strains from different salinity environments that we were maintaining in culture as they represented isolates from distinctly different salinity environments that represented two "end-member" marine salinity conditions. This reasoning will be added to the methods section in the revised manuscript.

**Revisions made:** We have added a brief explanation of our choice to the Methods (lines 103-105).

- I find the data produced in this work very useful for the scientific community working on the interpretation of changes of coccolith morphometrics in relationship with environmental variability. However, as often different strains exhibit differing response to a changing environmental parameter it is important that the reader is able to associate the cultured strains to specific *E. huxleyi* morphotypes (which are the forms they find in the field). For this reason, it would be useful for the reader that authors provide a description of the morphology of the coccoliths of each strain and also that they provide some SEM pictures (either in the main text or as supplementary materials). A general description of the main existing *E. huxleyi* morphotypes could be found here: (https://www.mikrotax.org/Nannotax3/index.php?taxon=Emiliania%20huxleyi&module=Coccolit hophores).

We agree that different strains can exhibit variable morphology (independent of any physiological or environmental influence) and that this is therefore important context for examining morphological changes under an environmental treatment. A detailed description of the differences in morphology between these two strains has already been presented in Gebühr et al. (2021), with illustrative images, but we agree it would be useful to provide a brief summary of the morphological characteristics (including differences in range of coccolith and coccosphere sizes) of each strain within the revised manuscript. We will also direct readers to Gebühr et al. (2021) for a more detailed description and for SEM images of the two strains showing these morphological features.

**Revisions made:** In the Discussion (lines 498-501), we now highlight that both strains are morphotype A and describe that RCC1232 is larger than PLYB11 overall. We also now refer the reader to a more extensive description of the strain differences previously published in Gebühr et al. (2021) and we have prepared a new supplementary figure (Fig. S1, uploaded to zenodo alongside the datasets accompanying this study) that shows SEM images of coccospheres under all experimental conditions for both strains, where the reader can see the morphology clearly.

- Authors mentioned in lines 344-345 that "Cellular PIC could not be estimated for RCC1232 under salinity conditions as coccospheres were too poorly preserved". Providing some images of this poor preservation would be also helpful for those researchers working on the field. Perhaps, similar forms have been documented in the natural environment.

We will add a plates of illustrative SEM images of poor preservation (e.g. collapsed coccospheres that prevented the measurement of coccosphere size) under salinity 25 conditions as a supplement to the revised manuscript.

**Revisions made:** The new supplementary figure (Fig. S1) now shows examples of poorly preserved coccospheres of RCC1232 under salinity 25 conditions alongside images from all other experiments at three timepoints through the experiments. We hope that this is a useful resource for those working in the field.

- Could authors explain a little more about how Ks values were estimated for the strains in the publications mentioned in lines 165-167 (just a couple of lines) and explain with value of the range provided for each strain was used in the present study?

We will add a brief description of the method used by Linge Johnsen et al. (2019) to derive the shape factors that we used. Thank you for highlighting that it is not clear how the range of Ks values stated in the manuscript have been applied, this will be clarified in the text of the revised manuscript. We apologise for the error in the +/- symbol used in the table caption and elsewhere that resulted during pdf conversion, this will be corrected in the revised manuscript.

**Revisions made:** We have added further explanation of the source of the $K_s$ values to the Methods and specified the exact $K_s$ value used for each strain under each salinity condition for clarity (lines 187-190).

- Line 350, heading of table 1. Could you please revise the symbol before sd? Do you mean + -?

**Revisions made:** We did mean +/- in the caption for Table 1 and unfortunately in the process of pdf conversion this resulted in an error. We have attempted to rectify this in the revised manuscript and hope that this error does not occur again.

- Lastly, it would be useful for the readers that authors mention somewhere in the text what the expected changes in salinity are for the global ocean and if the range of salinity values tested in this study fall within the expected values of salinity change.

Current projections for changes in global ocean salinity will be added to the introduction (see also response to reviewer 1) and we agree that this is useful context for the motivation of the study. We will also add to the methods how the choice of salinity treatment conditions (25 and 45) relates to the range of present-day and projected future ocean salinity conditions for additional context.

**Revisions made:** As previously mentioned for a similar comment raised by Reviewer 1, we have added a brief description of the expected changes in global ocean salinity to the Introduction (lines 42-46). We have also provided justification for the range of applied salinity treatments to the Methods (lines 115-120) and provided additional literature support of the magnitude of salinity change experienced by marine organisms in the Introduction.